# Cell environment shapes TDP-43 function with implications in neuronal and muscle disease

Urša Šušnjar[1], Neva Škrabar[2,3], Anna-Leigh Brown[4], Yasmine Abbassi[1], Hemali Phatnani[5], NYGC ALS Consortium*, Andrea Cortese[4,6], Cristina Cereda[7], Enrico Bugiardini[4], Rosanna Cardani[8], Giovanni Meola[9,10], Michela Ripolone[11], Maurizio Moggio[11], Maurizio Romano [12], Maria Secrier [13], Pietro Fratta [4] & Emanuele Buratti [1]✉

TDP-43 (TAR DNA-binding protein 43) aggregation and redistribution are recognised as a hallmark of amyotrophic lateral sclerosis and frontotemporal dementia. As TDP-43 inclusions have recently been described in the muscle of inclusion body myositis patients, this highlights the need to understand the role of TDP-43 beyond the central nervous system. Using RNA-seq, we directly compare TDP-43-mediated RNA processing in muscle (C2C12) and neuronal (NSC34) mouse cells. TDP-43 displays a cell-type-characteristic behaviour targeting unique transcripts in each cell-type, which is due to characteristic expression of RNA-binding proteins, that influence TDP-43's performance and define cell-type specific splicing. Among splicing events commonly dysregulated in both cell lines, we identify some that are TDP-43-dependent also in human cells. Inclusion levels of these alternative exons are altered in tissues of patients suffering from FTLD and IBM. We therefore propose that TDP-43 dysfunction contributes to disease development either in a common or a tissue-specific manner.

[1] Molecular Pathology Lab, International Centre for Genetic Engineering and Biotechnology (ICGEB), Trieste, Italy. [2] Tumour Virology Lab, International Centre for Genetic Engineering and Biotechnology (ICGEB), Trieste, Italy. [3] Generatio GmbH, Center for Animal, Genetics, Tübingen, Germany. [4] Department of Neuromuscular Diseases, UCL Queen Square Institute of Neurology, London, UK. [5] Center for Genomics of Neurodegenerative Disease, New York Genome Center, New York, USA. [6] Department of Brain and Behaviour Sciences, University of Pavia, Pavia, Italy. [7] Genomic and post-Genomic Unit, IRCCS Mondino Foundation, Pavia, Italy. [8] BioCor Biobank, UOC SMEL-1 of Clinical Pathology, IRCCS-Policlinico San Donato, San Donato Milanese, Italy. [9] Department of Biomedical Sciences for Health, University of Milan, Milan, Italy. [10] Department of Neurorehabilitation Sciences, Casa di Cura del Policlinico, Milan, Italy. [11] Neuromuscular and Rare Diseases Unit, Department of Neuroscience, Fondazione IRCCS Ca' Granda Ospedale Maggiore Policlinico, Milan, Italy. [12] Department of Life Sciences, University of Trieste, Trieste, Italy. [13] UCL Genetics Institute, Department of Genetics, Evolution and Environment, University College London, London, UK. *A list of authors and their affiliations appears at the end of the paper. ✉email: buratti@icgeb.org

TDP-43 (TAR DNA-binding protein 43), a protein encoded by the *TARDBP* gene, is a ubiquitously expressed member of heterogeneous nuclear ribonucleoprotein (hnRNP) family able to bind DNA and RNA that participates in various steps of mRNA metabolism, including transcription, pre-mRNA splicing, miRNA generation, regulation of mRNA stability, nucleo-cytoplasmic transport and translation[1–4]. TDP-43 was initially described as the major component of cytoplasmic inclusions formed in motor neurons of patients suffering from amyotrophic lateral sclerosis (ALS) and frontotemporal dementia (FTLD) despite the fact that mutations in *TARDBP* gene only account for a small subset of those cases[5–7]. However, TDP-43 aggregates have also been found in skeletal muscles of patients with inclusion body myositis (IBM)[8,9], oculopharyngeal muscular dystrophy (OPMD)[10] and limb-girdle muscular dystrophy type 2a (LGMD2a)[11] suggesting that TDP-43 aggregation may play a prominent pathological role also in muscle tissue. Accordingly, TDP-43 myogranules have been shown to provide essential functions during skeletal muscle development and regeneration, both in mouse and human[12]. Despite ubiquitous expression of TDP-43, however, most studies investigating this protein have focused on its role in the central nervous system. Nonetheless, given its importance of TDP-43, both in muscle development and potentially in the pathogenesis of numerous myopathies, we have now systematically investigated functions elicited by TDP-43 in muscle (C2C12) and neuronal (NSC34) mouse cells in parallel.

Performing such a comparison is particularly interesting as these two cell environments display tissue-characteristic features, like for example: distinct post-translational modifications (PTMs) and cleavage products of TDP-43 described in muscles and neurons[13], muscle-characteristic localization of TDP-43 in space and time[12], cell-type-specific milieu of TDP-43 binding partners[14], and differential expression of RNA-binding proteins (RBPs) controlling common mRNA targets[15,16]. It is important to note that all these differences occur in a context of highly variable transcriptome between tissues including non-coding transcripts[17–19]. Therefore, TDP-43 might likely elicit tissue characteristic functions by targeting unique subsets of transcripts, which encode proteins participating in tissue-specific cellular pathways and provide crucial structural and functional features of a cell. The consequences of TDP-43 dysfunction in muscles could thus possibly differ from those that have so far been described in the central nervous tissue[20,21].

In the last decade, high throughput methodologies have shifted the focus from characterization of individual events towards less biased global approaches, setting the ground for a systematic comparison of TDP-43 targeted RNAs across tissues and conditions. However, the overlap of TDP-43-controlled events identified by earlier studies is rather poor. It probably reflects the variation in technical approaches (microarrays, RNA-seq, CLIP-seq) and models employed in those studies: mouse brain[20], human post-mortem brain samples[21,22], human neuroblastoma cell line SH-SY5Y[21,23], HEK-293[24,25], Hela[26]. A clearer understanding of the extent to which TDP-43-mediated events are conserved between mouse and human is still lacking, yet it is a crucial point that should be addressed in future as it will allow better comparisons of human and mouse models of disease.

To finally address this issue in a systematic manner, we have identified subsets of unique cell-type-specific mRNA targets, as well as commonly regulated mRNAs, the tight regulation of which might underlie functions crucial for cell survival. More specifically, we have further explored splicing events that commonly occur in C2C12 and NSC34 cells and are additionally conserved in humans. We finally show that inclusion of common mouse-human TDP-43-regulated alternative exons is indeed altered in skeletal muscles of IBM patients and different

brain regions of ALS and FTLD patients with reported TDP-43 pathology.

## Results

**TDP-43 expression is similar in C2C12 and NSC34 cells.** To start comparing the functions of TDP-43 in cells of muscular and neuronal origin, we used the most commonly employed mouse cell lines representing skeletal muscle (C2C12) and motor neurons (NSC34). They have been previously used to study TDP-43-associated neurodegeneration as well as the role of TDP-43 in muscle development[12,27–29]. We first assessed protein levels of endogenous TDP-43 in untreated cells (Fig. 1a). Although in mature mouse tissues TDP-43 expression was reported to be higher in the brain compared to quadriceps muscle[30], we noted no difference in the amount of total TDP-43 protein between undifferentiated C2C12 and NSC34 cells (Fig. 1a), nor in the expression of TDP-43 at the RNA level of siLUC-transfected cells (Fig. 1b, Supplementary Data 1).

TDP-43 was silenced to a similar extent in both cell lines (Fig. 1b, Supplementary Data 1), and reduction of the protein was confirmed by western blot (Fig. 1c). TDP-43 loss functionally reflected in altered splicing of the two well characterized target transcripts *Poldip3* and *Sort1* (Fig. 1d)[23,26,31,32]. To explore transcriptome-wide effects of TDP-43 downregulation, we then performed RNA-seq analysis on polyadenylated mRNA extracted from TDP-43 depleted cells. Both cell lines displayed a characteristic transcriptome as revealed by principal component analysis (PC1), whereas the effect of TDP-43 knockdown explained a smaller portion of the variation between samples (PC2) (Fig. 1e, Supplementary Data 1). This result suggests that TDP-43 silencing promotes alterations in overall mRNA abundance in C2C12 and NSC34 based on the tissue-characteristic transcriptome.

**mRNAs dysregulation following TDP-43 reduction in C2C12 and NSC34 cells is cell-type specific.** Tissues substantially vary with regards to abundance of transcripts encoded by individual genes and splice isoforms they express, and these differences underlie specific biological characteristics and functions. To examine the effect of TDP-43 loss on expression levels (differential gene expression, DEG) in two cell lines, we separately normalized reads of C2C12 and NSC34 datasets and obtained 4019 transcripts, expression levels of which were subject to TDP-43 regulation. At $p_{adj} < 0.05$, we detected a very similar number of DEG in C2C12 and NSC34 (2325 and 2324, respectively), with 630 (15.7%) transcripts being commonly dysregulated in both cell lines (Fig. 2a, Supplementary Data 2). Surprisingly enough, the small overlap could not be explained by the fact that some genes are expressed in a cell-type-specific manner (i.e., muscle-characteristic genes are not transcribed in neuronal cells and vice versa), as the overlap between TDP-43 targets remained small (19.3%) even if we only considered genes expressed in both cell lines (FPKM in both cell lines > 0.5) (Fig. 2b, Supplementary Data 1 and 2). However, our data indicated that TDP-43 targets regulated in a cell-type-specific fashion are highly expressed in one cell type but not in the other. On average, C2C12-specific TDP-43-regulated mRNAs show higher expression in C2C12 than in NSC34 cells and vice versa (Supplementary Fig. 1a).

It has previously been proposed that TDP-43 binding is needed to sustain pre-mRNA levels and that mRNA downregulation would be a direct consequence of TDP-43 loss, while mRNA upregulation was explained by indirect effects[20]. In our datasets (Fig. 2a, Supplementary Data 2), the number of downregulated genes slightly outnumbered genes that were upregulated following TDP-43 depletion (Supplementary Fig. 1b); however, the overlap was very similar, irrespective the direction of the change

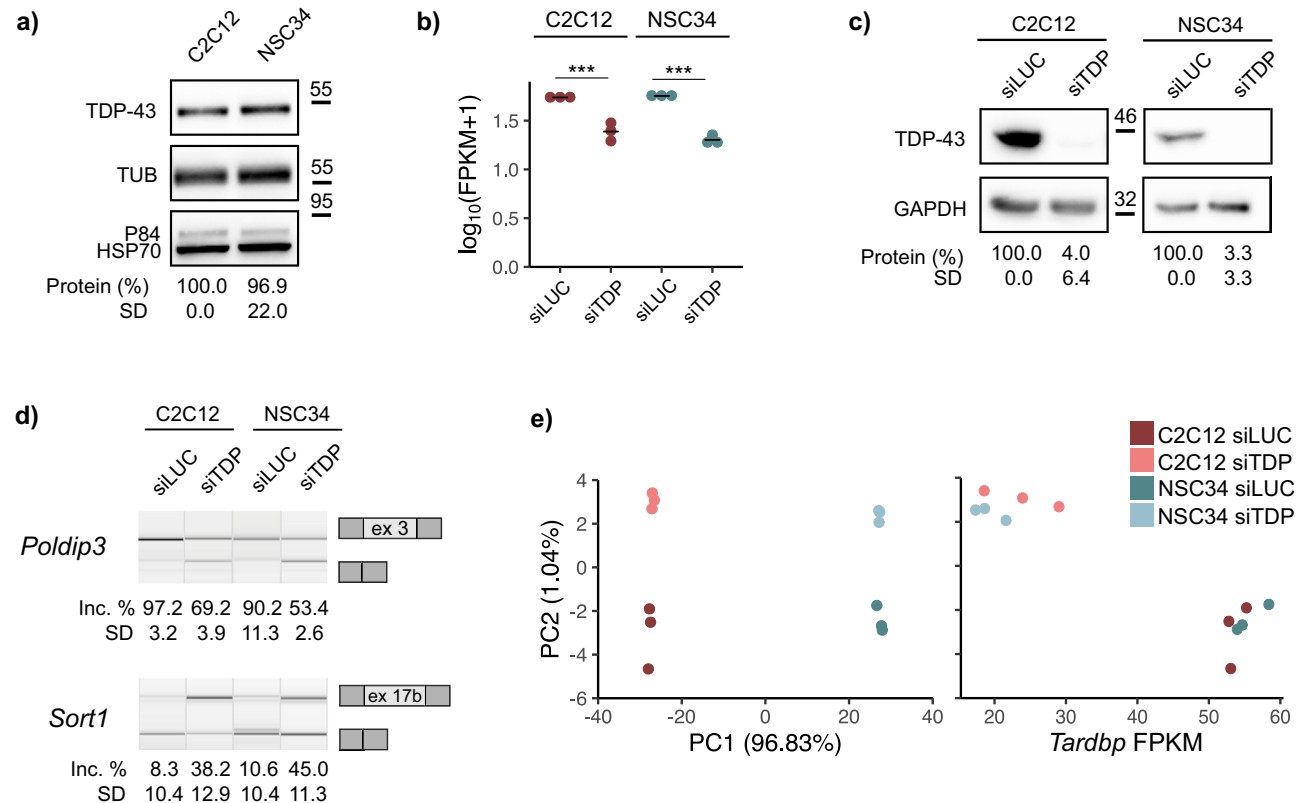

**Fig. 1 TDP-43 expression and functional consequences of TDP-43 silencing in C2C12 and NSC34 cells. a** Western blot shows similar expression of endogenous TDP-43 in C2C12 and NSC34 cells. The amount of TDP-43 was normalized to the sum of peak intensities of 3 loading controls (tubulin, HSP70 and P84) ($n = 3$ replicates per group). **b** Expression levels of *Tardbp* in TDP-43-silenced C2C12 and NSC34 and corresponding controls assessed by RNA-sew plotted as $\log_{10}$-transformed FPKM values show TDP-43 was depleted (on the mRNA level) to the same extent in both cell lines ($n = 3$ replicates per group). $p_{adj} = 1.6 \cdot 10^{-18}$ for C2C12 and $p_{adj} = 2.3 \cdot 10^{-72}$ for NSC34. *p*-values were generated using Wald test and Benjamini–Hochberg multiple testing correction[74]. **c** Western blot shows reduction of TDP-43 in C2C12 and NSC34 cells upon siTDP transfection. siLUC-transfected cells were used as a control. TDP-43 expression was normalized against GAPDH ($n = 3$ replicates per group). **d** TDP-43 depletion led to altered splicing of *Poldip3* and *Sort1*. Semi-quantitative RT-PCRs conducted in TDP-43-silenced samples and corresponding controls are shown along with the quantification of splicing changes (% of alternative exon inclusion). The number of the alternative exon is given below (see the exact transcript numbers in Supplementary Table 1, $n = 3$ replicates per group). **e** PCA plot visualizes distances between siLUC- and siTDP-transfected C2C12 and NSC34 cells based on FPKM of all genes obtained by RNA-seq (left). Variation in the PC2 is explained by the presence/absence of TDP-43 (right).

(14.0% and 15.0% for upregulated and downregulated transcripts, respectively). Comparing the extent of expression changes of commonly regulated transcripts (630) induced by TDP-43 reduction, we saw a positive correlation ($\rho = 0.77$, *p*-value < 0.001) between two cell lines, with a trend towards larger alterations in C2C12 (Fig. 2c, Supplementary Data 2). Of note, there were few mRNAs whose expression was altered in the opposite direction in two cell lines, indicating that TDP-43 loss can elicit contrary effects depending on the cellular environment. Looking at individual target transcripts (Fig. 2d, Supplementary Data 2, Supplementary Fig. 1c, d), we hypothesized that the biggest expression changes induced by TDP-43 loss occurred in highly abundant transcripts. However, plotting the size of the change ($\log_2$ fold change) against background expression levels (FPKM in siLUC-transfected cells) of all DEG revealed that there is in fact no correlation between the two (Supplementary Fig. 1e).

Taken together, these results support the idea that unique sets of transcripts controlled by TDP-43 in each cell type can only partially be explained by variable expression levels of cell-type-characteristic mRNAs across tissues. Factors other than expression levels as such thus influence TDP-43 function that seems to be cell-type-specific. At the sequence level, in fact, TDP-43-regulated mRNAs detected in C2C12 or NSC34 appear to be equally well conserved across species (Supplementary Fig. 1f).

**Commonly enriched processes implicated in neurodegenerative and myodegenerative disease.** In mouse brain, TDP-43 has been shown crucial for maintenance of mRNAs that encode proteins involved in synaptic activity[20]. To elucidate which cellular processes might be controlled by TDP-43 in cells of muscle and neuronal origin, we conducted enrichment analysis of genes differentially expressed in C2C12 (2325) and NSC34 (2324) (Fig. 2e, Supplementary Data 5). Among C2C12 enriched GO terms, we found those directly associated with muscle characteristic features like *striated muscle development* or *muscle cell migration*, in line with results highlighting the importance of TDP-43 in skeletal muscle formation and regeneration[12,29]. On the other hand, a great portion of neuronal processes like *vesicle-mediated transport in synapse* or *regulation of postsynaptic membrane neurotransmitters* appeared to be affected by TDP-43 loss in NSC34 cells.

While the percentage of overlapping DEG was only 15.7%, by GO categories, almost a third of all biological processes (28%) enriched in C2C12 or NSC34 DEG (Fig. 2e, Supplementary Data 5) was commonly dysregulated upon TDP-43 depletion in both cell lines. Given that currently proposed picture of pathological processes implicated in myopathies bears several similarities with neurodegenerative disease[9,33,34], we investigated commonly enriched GO terms to see if any of them could detect

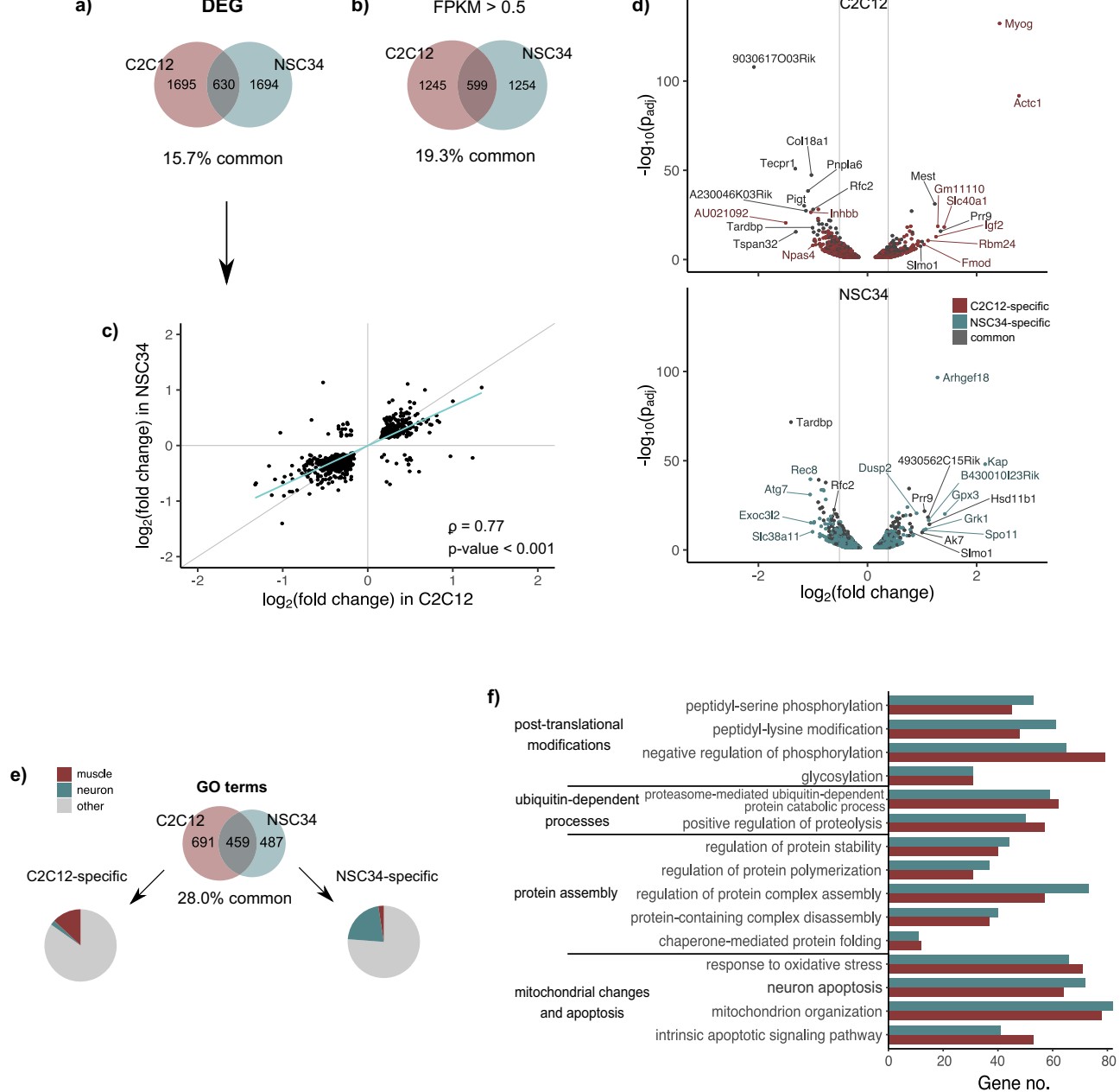

**Fig. 2 TDP-43 mediates expression levels of different mRNAs in C2C12 and NSC34 cells. a** Venn diagram shows the number of TDP-43-regulated transcripts identified in C2C12 and NSC34 cells exclusively (1695 and 1694, respectively), along with those that are commonly regulated by TDP-43 in both cell types (630). Transcripts with $p_{adj} < 0.05$ were considered as differentially expressed irrespective of their $\log_2$fold change. **b** Venn diagram shows the overlap (599 transcripts, 19.3%) of TDP-43-regulated DEG identified in C2C12 and NSC34 cell line (as in **a**, Supplementary Data 2), considering only transcripts expressed in both cell lines (FPKM in both cell lines > 0.5). Transcripts with $p_{adj} < 0.05$ were considered as differentially expressed irrespective of their $\log_2$-fold change. **c** Expression changes of common targets (**a**, Supplementary Data 2, 630) are plotted by their $\log_2$-fold change values in C2C12 and NSC34 (Spearman's $\rho = 0.77$, p-value < $2.2 \cdot 10^{-16}$). Grey line represents $y = x$ and the blue line represents the fitted regression. **d** TDP-43-mediated expression changes in C2C12 and NSC34 represented as volcano plots. C2C12- and NSC34-specific targets are shown in red and blue, respectively, while common targets are plotted as grey dots. Vertical lines indicate fold changes of 0.7 (30% increase) and 1.3 (30% decrease). Best hits are labelled with gene names. **e** Venn diagram shows the number of cell-type-specific and overlapping GO terms enriched by DEG identified in C2C12 or NSC34 cells. GO terms (category: biological process) were grouped based on their names as those implying muscle- (red) or neuron-related features (blue). **f** Representative GO terms (category: biological process) commonly enriched by DEG in C2C12 and NSC34 cells suggesting pathological abnormalities described in neurodegenerative and myodegenerative disease (hand curated).

abnormalities previously described in the above-mentioned diseases. Significant GO terms enriched by DEG in both C2C12 and NSC34 (Fig. 2f, Supplementary Data 5) suggest that some common TDP-43-mediated mechanisms might contribute to development of TDP-43-proteinopathies in both muscle and

neuronal tissues. Pathomechanisms include aberrant protein accumulation (i.e., of ubiquitin, amyloid β, α-synuclein, phosphorylated τ and TDP-43), post-translational modifications of deposited proteins (phosphorylation, ubiquitination, acetylation, sumoylation), defects in protein disposal (26 S proteasome and

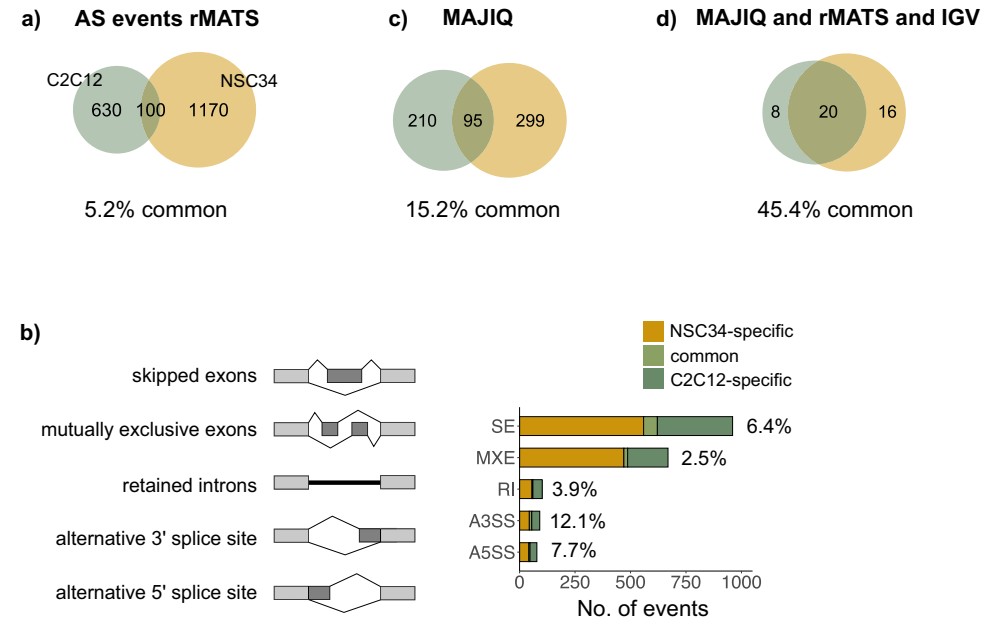

**Fig. 3 TDP-43-regulated splicing changes show cell-type specificity. a** Venn diagram shows the total number of AS events (detected by rMATS at FDR < 0.01) induced by TDP-43 depletion in C2C12 and NSC34 specifically (630 and 1170, respectively), together with those commonly detected in both cell lines (100). **b** The number of annotated AS events (Fig. 3a, Supplementary Data 3) visualized by event type. SE - exon skipping, MXE - mutually exclusive exons, RI - intron retention, A3'SS and A5'SS - alternative 3' or 5' splice site. The percentage of overlapping AS events is reported on the plot. **c** Venn diagram shows the number of AS events (junctions) detected by MAJIQ (ΔPSI > 0.2, FDR < 0.1, Supplementary Data 4). **d** Venn diagram showing the number of AS events detected by rMATS (Fig. 3 a, Supplementary Data 3) and MAJIQ (Fig. 3b, Supplementary Data 3), that were additionally validated using Interactive Genome Viewer.

autophagy) and mitochondrial abnormalities. However, while there was a greater overlap between biological response to TDP-43 depletion (GO: biological process), the specific differentially expressed transcripts in common terms were remarkably different between C2C12 and NSC34 (Supplementary Fig. 1g). This implies that TDP-43 can influence similar biological processes in both muscles and neurons, but it does so by mediating expression levels of genes encoding for distinct proteins that participate in those pathways.

**TDP-43-mediated splicing is more pronounced in NSC34 cells.** Along with mRNA depletion, aberrant pre-mRNA splicing has been described to contribute to neuronal vulnerability as a consequence of pathologic TDP-43 behaviour[20,21,35]. Yet, little is understood about how TDP-43 dysfunction affects pre-mRNA splicing in tissues beyond the central nervous system. In this work, we systematically compared alternative splicing (AS) alterations following TDP-43 reduction in C2C12 and NSC34 cells. As expected, a considerably lower number of splicing events was detected in C2C12 than in NSC34 cells (730 and 1270, respectively) using rMATS at FDR of 0.01 (Fig. 3a, Supplementary Data 3), which held true for events of any classical AS category (i.e., SE, MXE, RI, A3'SS, A5'SS) (Fig. 3b, Supplementary Data 3). Neuronal and muscle targets did not vary with regard to event type proportion (Supplementary Fig. 2a); length of cassette exons (Supplementary Fig. 2b); the ratio between inclusion/exclusion events (Supplementary Fig. 2c); or percentage of frame-conserving events (Supplementary Fig. 2d). Interestingly enough, alternative sequences regulated by TDP-43 in the neuronal cell line seem to be more conserved across species than TDP-43-regulated sequences in muscle cell line (Supplementary Fig. 2e). This holds true particularly for cassette exons (Supplementary Fig. 2f), which represent the most frequent event type detected by our pipeline (Supplementary Fig. 2a).

The observation that TDP-43 regulates more events in NSC34 cells might reflect the importance of alternative splicing as a regulatory mechanism in neurons and support the existence of a distinct splicing programme in neuronal tissues, as already suggested by others[14,36,37]. Moreover, very few AS events detected by rMATS (on average 5.2%) appear to be commonly regulated by TDP-43 in both cell types, with the percentage of overlapping AS events being small (5.8%) even when we only considered AS in transcripts commonly expressed in both cell lines (FPKM > 0.5) (Supplementary Fig. 2g).

Even though Jeong et al.[30] have previously reported that TDP-43's repression of cryptic exons is tissue-specific, the percentage of overlapping regular (annotated) AS events seemed unexpectedly low. We therefore conducted an independent analysis using an alternative splicing tool—MAJIQ[38], (ΔPSI > 0.2, FDR < 0.1) (Fig. 3c, Supplementary Data 4). Like with rMATS, a higher number of TDP-43-dependent splicing events was detected in NSC34 cells (305 and 394 in C2C12 and NSC34, respectively). MAJIQ and rMATS quantify in separate ways (junctions or events, respectively)[39]; thus, comparable results are only produced when the same pipeline is applied. The proportion of commonly regulated annotated splicing events obtained by MAJIQ relative to rMATS (15.5% and 5.2%, respectively) clearly points out to the different performance of two event-based methods on the same dataset[39].

To answer the question how cell-type-specific the activity of TDP-43 really is, we, as our final approach, estimated the overlap of TDP-43-controlled splicing between C2C12 and NSC34 cell line on a smaller subset of AS events, all of which were detected by both splicing tools (even if at the expense of false negatives). Moreover, we visually validated every event found by splicing tools using Interactive Genome Viewer and subsequently excluded those whose changes could not be confirmed (Fig. 3d, Supplementary Data 3 and 4).

Amongst these events, 45% of TDP-43-regulated splicing activity seemed to be shared between NSC34 and C2C12, and we

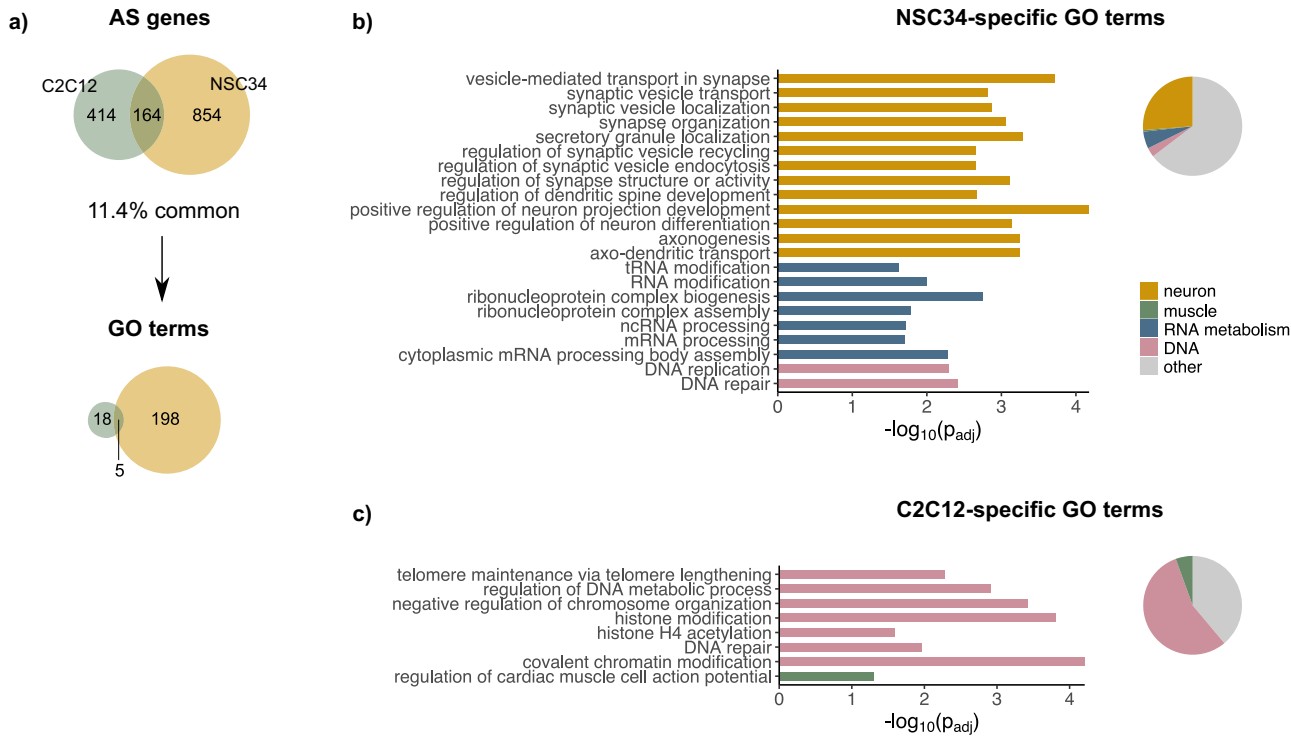

**Fig. 4 TDP-43-regulated transcripts are enriched in neuronal functions and DNA-related processes. a** Venn diagrams show the number of alternatively spliced transcripts (as detected by rMATS at FDR < 0.01 (Fig. 3a, Supplementary Data 3)) in C2C12 and NSC34 cells together with GO terms (category: biological process, $p_{adj}$ < 0.05) enriched in AS genes detected in each cell line. **b** GO terms uniquely enriched in NSC34 (198) imply on deregulation of neuronal processes, mRNA metabolism and DNA biology in NSC34 cells (representative GO terms are shown on the plot). **c** GO terms uniquely enriched in C2C12 (18) suggest involvement of TDP-43-regulated AS genes in DNA-modifying processes (representative GO terms are shown on the plot).

firmly believe this provides a better estimate of TDP-43's cell-type-specificity.

**Alternatively spliced TDP-43 targets are implicated in neuronal functions and DNA-related processes**. We further employed GO analysis to see whether genes with TDP-43-regulated splicing identified in C2C12 and NSC34 form interconnected networks and if TDP-43 can, by mediating AS, influence particular biological processes in each cell type. Since the number of C2C12 AS genes entering GO analysis (578) was considerably lower than that of NSC34 genes (1018), the analysis resulted in fewer GO terms found to be enriched in C2C12 compared to many in NSC34 (23 and 203, respectively) (Fig. 4a, Supplementary Data 3 and 5). As expected, GO terms enriched in NSC34 cells exclusively suggest that in these cells, alternatively spliced mRNA predominantly encode for proteins implicated in processes taking place in the nervous system (e.g., *axonogenesis, regulation of neuron differentiation*) (Fig. 4b, Supplementary Data 5). This is in line with earlier studies, which demonstrated that in human neuroblastoma cells SH-SY5Y TDP-43-dependent splice isoforms encode for proteins regulating neuronal development and those involved in neurodegenerative disease[21].

On the other hand, GO terms (56%) enriched in C2C12 cells exclusively (Fig. 4c, Supplementary Data 5) suggested involvement of AS genes in DNA-related processes (e.g., *covalent chromatin modification* or *regulation of chromosome organization*), while only one implied a muscle characteristic feature (i.e., *regulation of cardiac muscle cell action potential*). As we thought this observation might be biased due to the low number of GO terms detected in C2C12 (18), we repeated enrichment analysis, this time using a more relaxed threshold (non-corrected *p*-value < 0.01 instead of FDR < 0.01) on AS genes that would enter

GO analysis. However, even among 45 enriched GO terms obtained using less stringent threshold, DNA-related processes comprised more than a third of all GO terms (36%, Supplementary Fig. 2h), which was not the case for NSC34 cells.

**RBPs expressed in NSC34 and C2C12 influence TDP-43-dependent splicing**. The observation that TDP-43 loss elicits a tissue-characteristic response did not come as a surprise, as RNA-binding proteins (RBPs) other than TDP-43 might be differentially expressed in these cells. Inspecting expression levels of some RNA-binding proteins[14], which either directly interact with TDP-43[40] or influence processing of its target transcripts[16,31,41], we saw a higher average expression of RBPs in neuronal NSC34 cells (Fig. 5a, Supplementary Data 1) in line with previous observations[14]. Their joint functions in coordinating mRNA processing might underlie a more complex splicing regulation that is unique for neuronal tissues and explain why TDP-43-regulated splicing is more frequent in NSC34 than in C2C12 cells (Fig. 3). The two cell types clearly express a distinct array of RBPs (Fig. 5b, Supplementary Data 1), while abundance of some mRNAs is additionally affected by TDP-43 depletion (Fig. 5c, Supplementary Data 2).

Having identified and validated the most representative cell-type-specific AS events (Fig. 6b), we next explored whether TDP-43-dependent expression of specific splicing isoforms could be explained by the co-regulatory activity of other RBPs that are highly expressed in one cell line but are absent (or have negligible levels) in the other. Using siRNA against these proteins, we knocked down four RBPs that are most prominently expressed in a cell-type-specific manner (i.e., *Elavl4* and *Elavl3* in NSC34; *Celf2* and *Khdrbs3* in C2C12) (Fig. 6a). As a consequence, we could see that in three events out of the six tested, the expression of tissue-characteristic TDP-43-regulated splice isoforms (Fig. 6b, c)

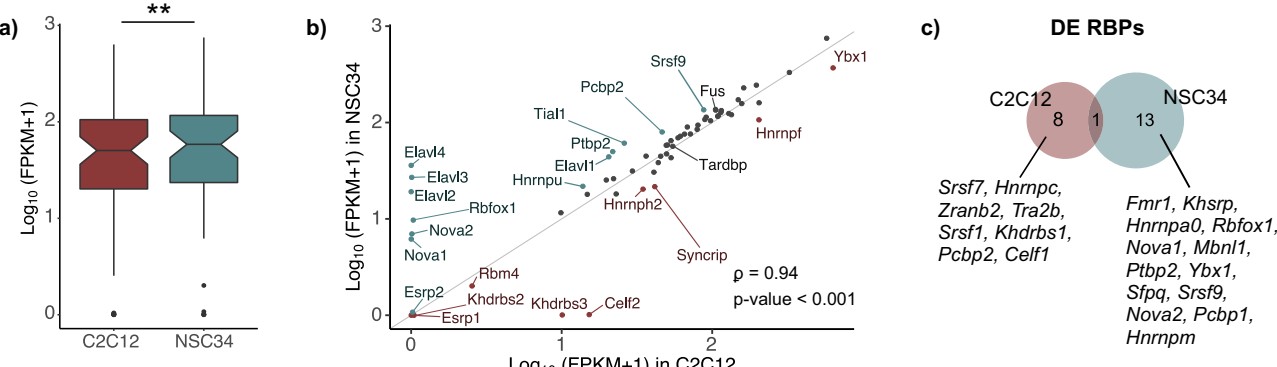

**Fig. 5 Expression of RNA-binding proteins in C2C12 and NSC34 cells. a** Boxplot shows that NSC34 cells on average display higher expression of 63 RNA-binding proteins[14] compared to C2C12 cells (*p*-value = 0.0028). Average expression levels are plotted as log₁₀-transformed FPKM values of all 63 transcripts and *p*-value was generated using Wilcoxon signed-rank test. **b** Expression of 63 RBPs (plotted as log₁₀-transformed FPKM values) in C2C12 and NSC34 cells (Spearman's $\rho$ = 0.94; *p*-value < 2.2·10⁻¹⁶). Those with higher expression in one cell line than another (> 150%) are shown in red (C2C12) or blue (NSC34). Grey line represents *y* = *x*. **c** Venn diagram shows RBPs the expression of which changes following TDP-43 reduction. The overlapping event is downregulation of *Tardbp*.

indeed depends on the co-regulation by these tissue-specific splicing actors. For example, by silencing muscle-characteristic RBPs *Celf2* or *Khdbrs3* in C2C12 cells, we decreased the expression of the long *Tbc1d1* isoform typical of C2C12 cells and obtained a more "neuron-like" phenotype even in the presence of TDP-43 (Fig. 6d). By silencing *Elavl3* or *Elavl4* in NSC34 cells, on the other hand, we could shift the ratio of *Dync1i2* and *Lrp8* splice isoforms expressed in NSC34 cells towards the isoform typically present in C2C12 cells (Fig. 6e). Although a more comprehensive analysis of the role played by all tissues-specific splicing factors (also in combination with one another) would be necessary to obtain a complete picture, these results show that the cell-type-specific splicing activity of TDP-43 arises as consequence of different cellular environments, and especially the characteristic expression of tissue-specific RBPs that modulate inclusion of selected TDP-43-targeted exons.

**Common TDP-43 splicing targets detected in C2C12 and NSC34.** Previous studies have already disclosed lists of transcripts, whose splicing is affected by TDP-43 removal or dysfunction[21,24,28,41]. Yet, the reproducibility of target identification is rather poor, possibly due to differences in methodological approaches, low conservation of TDP-43 targets across species[28], and, as we show, the unique function TDP-43 elicits in each tissue or celltype. The most consistently reported TDP-43-regulated splicing event across studies and conditions is skipping of exon 3 within *Poldip3*/*POLDIP3* mRNA (both mouse and human)[23]. This being so, inclusion level (percent spliced in, ΔPSI) of *Poldip3* exon 3 often serves as a readout of TDP-43 functionality[42–44]. In search of new splicing events that would, similarly to *Poldip3*/*POLDIP3*, show high reproducibility across experimental settings, we chose mRNAs that underwent the biggest shift in TDP-43-dependent exon inclusion and whose isoform proportion was altered in both C2C12 and NSC34 cells. The isoform switch of these targets was validated using isoform-sensitive semiquantitative RT-PCR (Fig. 7a).

Relative to cell-type-specific TDP-43 targets, commonly spliced transcripts on average show higher expression in C2C12 and NSC34 cells than transcripts alternatively spliced in a cell-type-specific manner (Fig. 7b, Supplementary Data 1). Furthermore, commonly detected events display bigger splicing transitions (bigger ΔPSI) (Fig. 7c, Supplementary Data 3). Most of the splicing changes detected in C2C12 and NSC34 occurred in the same direction (83%, $\rho$ = 0.62, *p*-value < 0.001) (Fig. 7d,

Supplementary Data 3), meaning that for that subset of transcripts, TDP-43 exerts a similar function in cells of neuronal and muscular background. We observed a higher frequency of frame-preservation among splicing events found to be controlled by TDP-43 in both cell lines (Fig. 7e, Supplementary Data 3) along with better conservation of common TDP-43-regulated sequences across species (Fig. 7f).

We further compared results obtained in TDP-43-silenced cells (NSC34 and C2C12) to an RNA-seq dataset previously generated by Fratta et al.[45], in which they sequenced brain tissue of mice carrying a mutation in important RNA-recognition motif RRM2 (F210I) of endogenous *Tardbp* gene leading to partial loss of TDP-43's splicing function (Supplementary Fig. 3).

That dataset was subjected to the same MAJIQ pipeline for detection of alternatively used splicing junctions (Fig. 3c, Supplementary Data 4). The overlap between splicing events detected in TDP-43-silenced NSC34 cells and brain tissue of mice with the RRM2 mutation was 10.1% compared to 15.2% of splicing events commonly identified in C2C12 and NSC34 following TDP-43 knockdown.

This may seem rather low, however, several considerations must be made: first, cell lines (C2C12 or NSC34, respectively) consist of homogenous cell populations, which is in contrast to heterogeneous cell composition within bulk brain tissue. Second, different from embryonic mouse brain (E18.5) containing neuronal cells at different stages of differentiation, only undifferentiated cells were used in our study (C2C12 and NSC34). It should also not be neglected that TDP-43 depletion in C2C12 and NSC34 cells was achieved by siRNA transfection whilst there was some functional TDP-43 still present in the F210I embryonic brain (as complete knockout of TDP-43 in mouse embryo leads to a very early embryonic lethality before day 12 of embryogenesis)[46].

We believe that results generated by a comparative analysis of TDP-43's function in NSC34 and C2C12 cells, that were chosen as two cell lines representing muscle and neurons, clearly demonstrate that the activity of TDP-43 itself depends on the cellular context. Nonetheless, NSC34 cells are used in undifferentiated state and can only to a limited extent recapitulate (structural and functional) features of mouse brain. This being said, we believe that a 10.1% overlap consistently supports the idea that there is a small subset of transcripts targeted by TDP-43 across different cell types (like for example *Poldip3*, *Ppfibp1* and *Tmem2*) while splicing of others strongly depends on cell-specific factors.

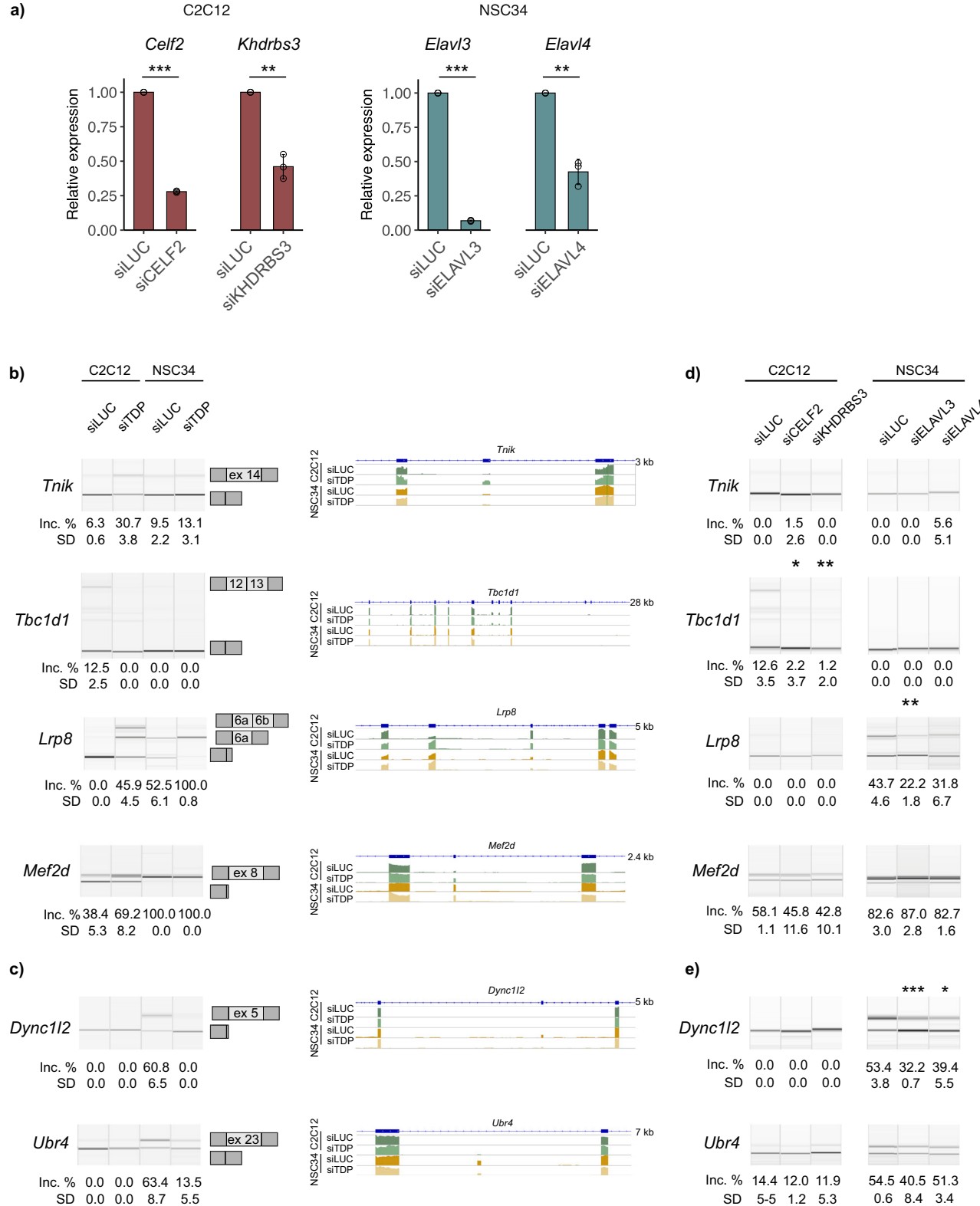

**TDP-43-regulated splicing events conserved between mouse and human**. While incorporation of exon 3 into mature *Poldip3* mRNA is regulated by TDP-43 in both mouse and human cells[23], most of TDP-43's regulated splicing has been shown to be highly species- and even cell-type-specific. We therefore investigated if any of commonly detected TDP-43 targets (Fig. 7a) are (according to VastDB[47]) predicted to have an orthologous event in humans. Some TDP-43-mediated events found in mice (*Rgp1*

exon 3, *Sapcd2* exon 2, *Fam220a* exon 2) do not even have a corresponding orthologous exon in humans. For those with putative AS orthology (i.e., the presence of orthologous alternative exon in both species), we tested whether alternative exons were subject to TDP-43 control also in human cells. We silenced TDP-43 in two human cell lines representing neuronal and muscular cells — human neuroblastoma SH-SY5Y and rhabdomyosarcoma RH-30[48] (Fig. 8a), which resulted in exon skipping

**Fig. 6 Cell-type-characteristic RBPs influence splicing of TDP-43 targets. a** Expression levels of four RBPs following their knock-down in C2C12 and NSC34 cells (*Celf2* and *Khdrbs3* were silenced in C2C12; *Elavl3* and *Elavl4* were silenced in NSC34) as assessed using qPCR. *p*-values were generated using Student's *t*-test (paired, two-tailed, *n* = 3 per group). **b** C2C12-specific and **c** NSC34-specific TDP-43-regulated splicing. Semi-quantitative RT-PCR conducted in TDP-43-silenced samples and corresponding controls is shown along with the quantification of splicing changes (% of alternative exon inclusion). The number of the alternative exon is shown in the scheme (see the exact transcript numbers in Supplementary Table 1). Coverage track from Interactive genome viewer (IGV) is shown on the right. **d** Splicing of C2C12-specific and **e** NSC34-specific TDP-43 targets depends on the presence of other tissue-specific factors. RT-PCR was conducted in samples silenced for specific RBPs (*Celf2* and *Khdrbs3* in C2C12; and *Elavl3* and *Elavl4* in NSC34 cells) together with corresponding controls. Quantification of splicing changes is shown below (% of alternative exon inclusion). Significance was tested using Student's *t*-test (paired, two-tailed, *n* = 3 per group).

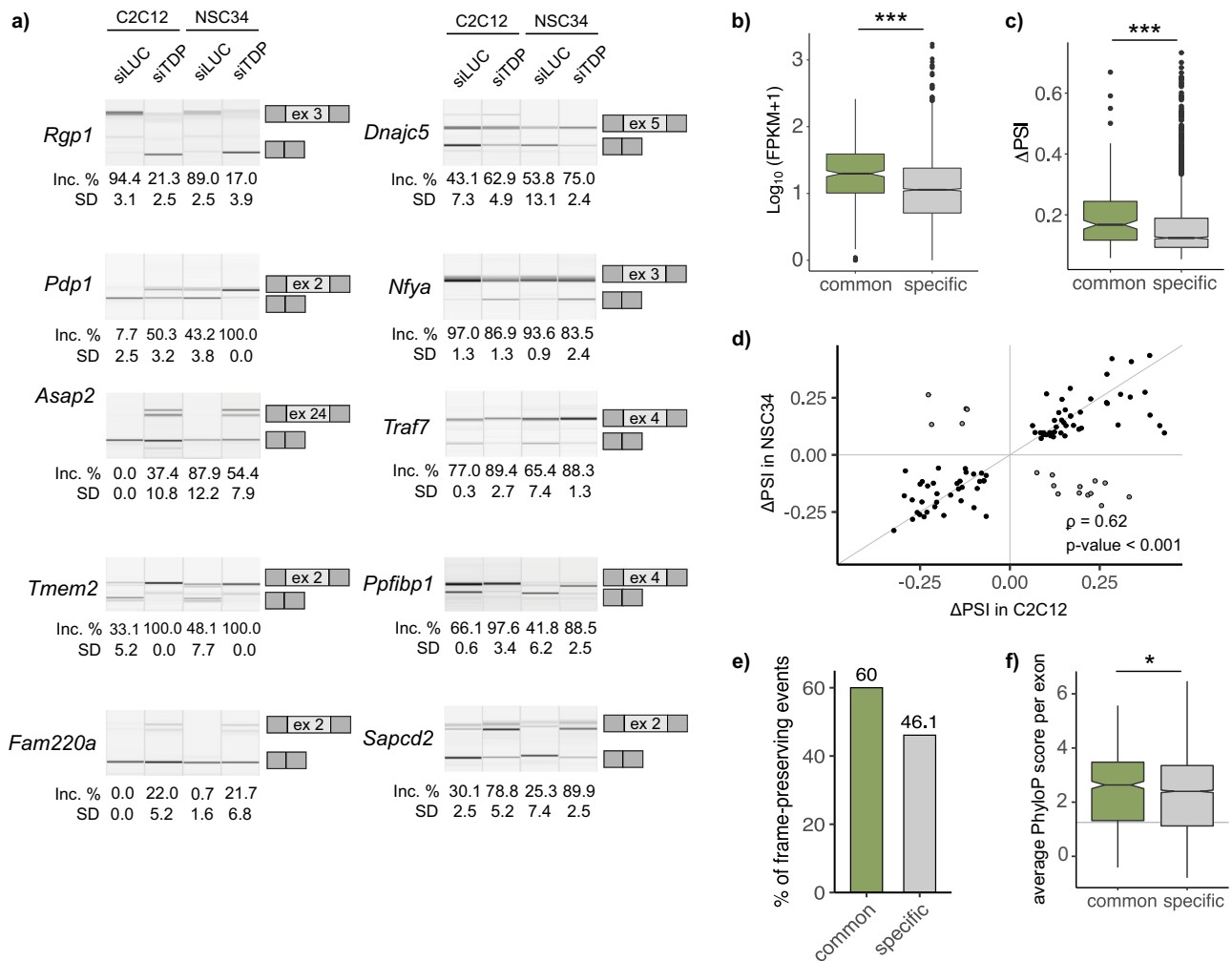

**Fig. 7 Commonly regulated TDP-43 splicing targets are more often frame-conserving, display higher expression levels and undergo bigger changes in isoform proportion. a** Validation of TDP-43 dependent splicing of 10 representative mRNA targets. Semi-quantitative RT-PCR conducted in TDP-43-silenced samples and corresponding controls is shown along with the quantification of splicing changes (% of alternative exon inclusion). The number of the alternative exon is shown in the scheme (see the exact transcript numbers in Supplementary Table 1, *n* = 3 replicates per group). **b** Average expression levels of transcripts that are commonly spliced in both cell lines (164) or in one cell line exclusively (1268) is plotted as $log_{10}$-transformed FPKM values (*p*-value < 2.2 · 10$^{-16}$). **c** Absolute changes (ΔPSI) of overlapping splicing events (100) compared to those uniquely occurring in C2C12 or NSC34 (1800) (*p*-value = 1.0 · 10$^{-7}$). *P*-values for **b**, **c** were generated by unpaired Wilcoxon rank-sum test. **d** The correlation of splicing changes for commonly detected splicing events (100) plotted as ΔPSI in C2C12 and NSC34 (Spearman's correlation coefficient $\rho$ = 0.62, *p*-value = 4.1 · 10$^{-12}$). **e** Percentage of frame-preserving AS events among those that commonly occur in both cell lines (100) and those regulated by TDP-43 in a cell-type specific manner (1800). **f** Average *per exon* PhyloP conservation scores plotted as box plots show TDP-43-regulated alternative sequences detected in both cell lines (634) are better conserved across species than those detected in one cell line exclusively (6019) (*p*-value = 0.02). *P*-value was generated using Wilcoxon rank-sum test, the grey line represents the median of average PhyloP scores of all exons in the mouse genome.

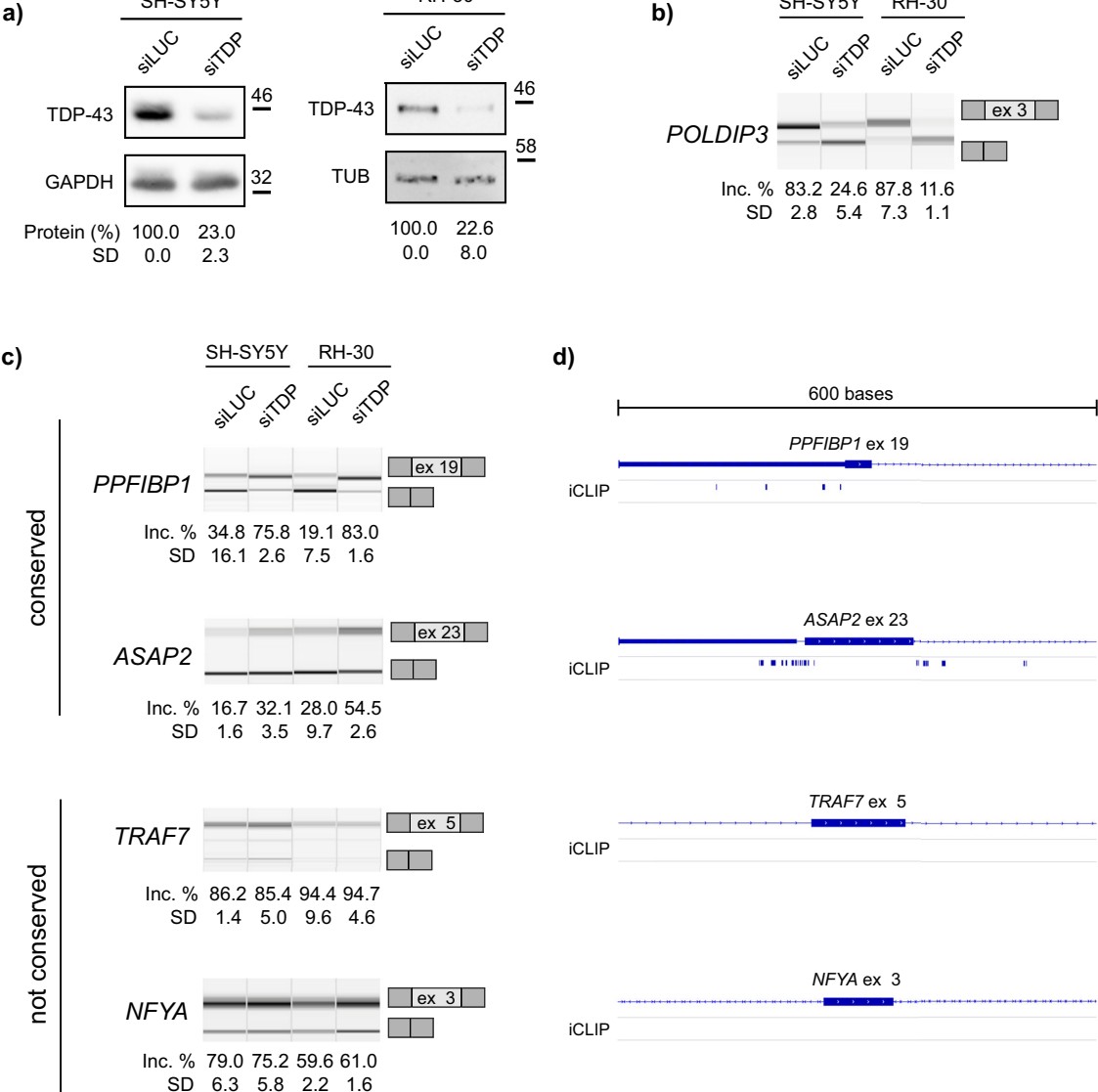

**Fig. 8 Alternative exons regulated by TDP-43 in mouse are subject to TDP-43 regulation in human cell lines or not. a** Western blot shows efficient reduction of TDP-43 in SH-SY5Y and RH-30 cells upon siTDP transfection. The amount of TDP-43 was normalized against GAPDH or tubulin ($n = 3$ replicates per group). **b** TDP-43 depletion led to altered splicing of *POLDIP3*. Semi-quantitative RT-PCR conducted in TDP-43-silenced samples and corresponding controls is shown along with the quantification of splicing changes (% of alternative exon inclusion). The number of the alternative exon is given below ($n = 3$ replicates per group). **c** Alternatively spliced exons regulated by TDP-43 in mouse cells are either subject to TDP-43 regulation in human cells (*PPFIBP1* exon 19 and *ASAP2* exon 23) or not (*TRAF7* exon 5 and *NFYA* exon 3). Semi-quantitative RT-PCRs conducted in TDP-43-silenced samples and corresponding controls are shown along with the quantification of splicing changes (% of alternative exon inclusion). Number of the alternative exon is given in the scheme (see the exact transcript numbers in Supplementary Table 2, $n = 3$ replicates per group). **d** Schematic representation of TDP-43 binding sites identified by iCLIP analysis in SH-SY5Y cells[21] in the vicinity of exons represented on **c**.

within *POLDIP3* (Fig. 8b). Likewise, TDP-43 depletion led to enhanced inclusion of exon 19 in *PPFIBP1* and exon 23 of *ASAP2* but not exon 5 of *TRAF7* or exon 3 of *NFYA* (Fig. 8c). Although one study reported a great portion of TDP-43-controlled exons in mouse to have a prior evidence of alternative splicing in humans[20] we still lack understanding to what extent TDP-43 regulation of mRNA processing is conserved between species. Exon orthology (as assessed by sequence similarity) could not be predictive of AS conservation since exon incorporation into mature mRNA depends on the exonic sequence but also on *cis*-regulatory motives and *trans*-acting factors[49–51]. In fact, iCLIP performed in SH-SY5Y cells[21] identified direct TDP-43-binding sites a in close proximity of alternatively spliced exons within *PPFIBP1* and *ASAP2*, while that was not the case for *TRAF7* and

*NFYA* (Fig. 8d). This finding suggests that alternative exons of *PPFIBP1* and *ASAP2* found to be regulated by TDP-43 in mouse and human cells are most likely controlled by TDP-43 in a direct fashion by its binding to regulatory sequences neighbouring splice sites.

**Altered splicing patterns imply on TDP-43 dysfunction in FTLD and IBM patients.** To explore if dysregulated alternative splicing could play a role in the pathophysiology of TDP-43 proteinopathies, we measured inclusion levels of TDP-43-mediated alternative exons in IBM muscles (Fig. 9a, Supplementary Data 6) as well as in pathological brain regions of ALS and FTLD cases with reported TDP-43 pathology (ALS-TDP and FTLD-TDP) (Fig. 9b, c,

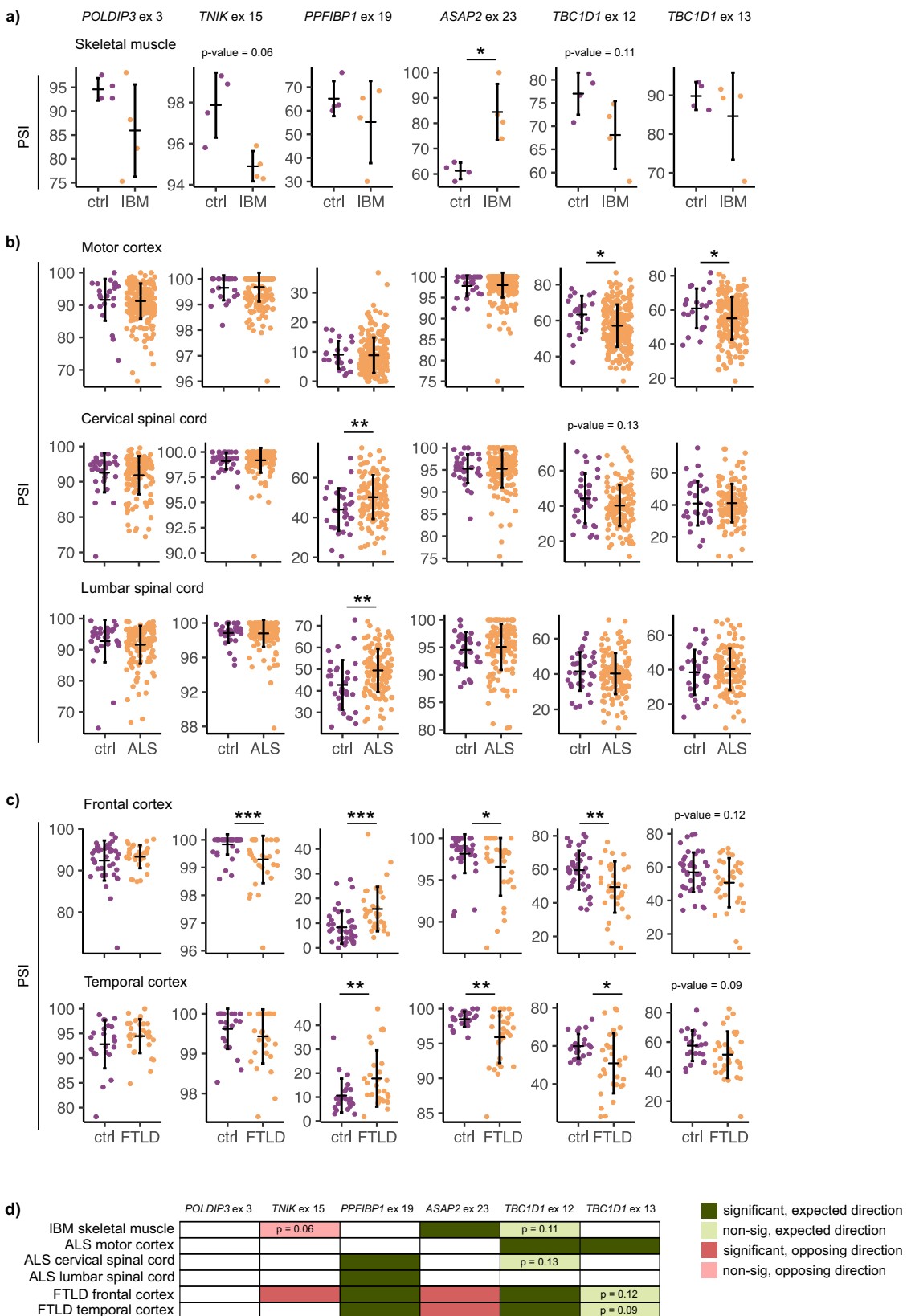

Supplementary Data 6). Since neuroanatomical regions markedly vary with regards to splice isoform expression (Supplementary Fig. 4a), we considered each brain region independently rather than analysing them together. Tissue-specific accumulation of truncated *STMN2*, which has recently been described as a very good clinical marker of TDP-43 impairment[22,43,52], in fact occurs in brain areas previously known to be affected by TDP-43 pathology. We thus investigated TDP-43-controlled splicing in the spinal cord (lumbar and cervical, respectively) and the motor cortex of ALS cases (Fig. 9b, Supplementary Data 6), whereas frontal and temporal cortices were the site of interest for FTLD patients (Fig. 9c, Supplementary Data 6).

**Fig. 9 Inclusion of TDP-43-controlled exons is altered in TDP-43-proteinopathies. a** Inclusion levels (PSI) of six alternative exons in skeletal muscle biopsies in IBM patients vs healthy controls ($n = 4$ per group). **b** PSI of six alternative exons in different brain regions (motor cortex, lumbar spinal cord, cervical spinal cord) of ALS patients and healthy controls (n motor cortex: 223 ALS and 23 ctrl, n cervical spinal cord: 134 ALS and 32 ctrl, n lumbar spinal cord 136 ALS and 33 ctrl). **c** PSI of six alternative exons in frontal and temporal cortices of FTLD patients with reported TDP-43 pathology and healthy controls (n frontal cortex: 33 FTLD and 40 ctrl, n temporal cortex: 30 FTLD and 23 ctrl). **a–c** P-values were generated using Wilcoxon rank-sum test. *p-value < 0.05, **p-value < 0.01, ***p-value < 0.001. **d** Schematic summary of all splicing alterations **a–c** detected in skeletal muscles of IBM patients and across neuroanatomical regions of ALS and FTLD patients compared to healthy controls. Dark green marks significant changes, which occur in the same direction as in TDP-43-depleted SH-SY5Y and RH-30 cells (refers to Fig. 6c); light green marks non-significant changes that occur in the expected direction (p-value is reported in the scheme); red marks significant changes occurring in the opposite direction relative to TDP-43-depleted SH-SY5Y and RH-30 cells; light red marks non-significant changes that occur in the opposite direction (p-value is reported in the scheme).

Splicing signature examined herein consisted of six TDP-43-regulated alternative exons: *POLDIP3* exon 3 is consistently detected as a TDP-43-regulated splicing event; exon 15 of *TNIK* has been previously described as TDP-43 target and was also detected in C2C12 (Fig. 6b), SH-SY5Y and RH-30 cells (Supplementary Fig. 4b); exon 19 of *PPFIBP1* and exon 23 of *ASAP2* are newly identified TDP-43 targets conserved across species; exons 12 and 13 of *TBC1D1* are detected to be controlled by TDP-43 in C2C12 cells and are associated with muscle differentiation. The long *TBC1D1* isoform, which is dependent on TDP-43, appears to be crucial in mature tissues but not in undifferentiated cells (hence, it was not detected in undifferentiated NSC34 (Fig. 6b), SH-SY5Y and RH-30 cells (Supplementary Fig. 4b)). As TDP-43 binding sites were indeed found in the proximity of these alternative exons (Supplementary Fig. 4c), inclusion levels of exons 12 and 13 of *TBC1D1* gene were investigated in mature tissues coming from patients.

Interestingly, when we assessed inclusion levels of six TDP-43-controlled AS events in patient tissues, we got distinct patterns. For example, out of the six AS events, we only observed increased *ASAP2* exon 23 inclusion in IBM muscle relative to healthy controls (Fig. 9a, Supplementary Data 6). In ALS cases, we detected significantly different inclusion of one exon (exon 19 of *PPFIBP1*) in the lumbar and cervical spinal cord but not in the motor cortex, while in motor cortex, we saw enhanced skipping of both alternative exons within *TBC1D1* (Fig. 9b, Supplementary Data 6). Intriguingly, FTLD appears to be the disease in which splicing of six alternative exons is most heavily perturbed. Multiple TDP-43-targeted exons show significantly altered inclusion in patients, both in frontal and temporal cortices (Fig. 9c, Supplementary Data 6). Apart from that, some non-significant changes clearly show a trend towards altered exon inclusion in patients.

At this point it is important to consider cell-type-specific splicing activity of TDP-43 (Fig. 3a, Supplementary Data 3), which makes it unlikely that upon TDP-43 malfunction, splicing of the same transcripts would be altered across all cell-types (Fig. 9d). This being said, the scheme in Fig. 9d summarizes splicing changes of six TDP-43-controlled exons detected in different tissues affected with TDP-43 pathology. The fact that splicing changes do not necessarily occur in the same direction following TDP-43 depletion in cell lines (as in the case of *ASAP2* and *TNIK*) again highlights the complexity of splicing control provided by TDP-43 that is generally acting within an interwoven network of splicing regulators. The same phenomenon (i.e., different directionality) was in fact observed when comparing the consequences TDP-43 depletion has on gene expression and alternative splicing in vitro using cell lines (Figs. 2c and 7d, Supplementary Data 2 and 3).

## Discussion
TDP-43 inclusions represent the hallmark of ALS/FTLD[5,7] and are frequently recognized as a secondary pathology in other neurodegenerative disease[53,54]. In recent years, great progress has been made in explaining how potential loss- and gain-of-function mechanisms contribute to the pathogenesis observed in the brain and spinal cord[27,45,55]. Nonetheless, a growing evidence of TDP-43 mis-localization and aggregation in tissues beyond the CNS has raised the possibility that TDP-43 dysfunction and consequently, impairment of RNA processing, might be deleterious for other tissues[42,56].

To this date, cell- and tissue-characteristic molecular features of TDP-43 have seldom been investigated in parallel. Considering recent attention that TDP-43 has received in IBM and related pathologies[8,9,11,57], we, therefore, sought to fill this gap. The purpose of our study has been to further explicate the role of TDP-43 in different tissues to better understand its involvement in pathogenesis in cell-types other than neurons, and to set the ground for development of potential therapeutic or biomarker strategies that focus on shared or specific disease mechanisms. We thus aimed to model protein loss/dysfunction in skeletal muscles vs. neurons and to focus on TDP-43-controlled alternative splicing (AS) events, as this is one of the best characterized features of this protein to date.

Although protein levels of TDP-43 itself are not different between C2C12 and NSC34, there is a tissue-characteristic expression of other RNA-binding proteins (e.g., those from *Elavl*, *Nova* and *Celf* families) that, like TDP-43, mediate RNA-related processes in a coordinated fashion. This result, together with differences in tissue-specific gene expression levels, can explain why there is little consistency across studies in identifying TDP-43-targeted transcripts[58], and it clearly outlines the importance of cellular context in shaping the functional role of TDP-43. With regards to future TDP-43 investigations, our findings highlight the need to employ tissue models, which are most relevant for a certain condition. Most importantly, our results show that in the case of TDP-43 proteinopathies, the knowledge acquired by studying neuronal cells could be translated to muscles only to a limited extent. Despite not being investigated in this work, the same presumably applies for the interpretation of iCLIP results, in which TDP-43 binding should be always considered in the context of tissue characteristic environment, having in mind possible differences in binding behaviour of the protein across cell-types that might affect splicing[59] and expression changes[43].

This initial comparison of TDP-43's function across cell-types based on RNA-seq discusses changes that occur at the mRNA level. With regards to mRNA splicing, it remains unclear whether TDP-43-dependent changes in mRNA isoforms actually translate in corresponding protein isoforms. This would require an extensive screening using isoform-specific antibodies or a mass-spectrometry-based approach. Nonetheless, as a proof-of-principle, we show that for TBC1D1, TDP-43-dependent ratio of two protein isoforms actually reflects that of *Tbc1d1* mRNAs (Supplementary Fig. 5).

In some cases, our parallel study has given expected results. In NSC34 cells, for example, TDP-43 loss impacts expression of

genes participating in pathways that provide elemental functions of neuronal cells, like vesicle-mediated transport and regulation of postsynaptic membrane neurotransmitters, which is perfectly in line with previous studies[20,21]. Similarly, the loss of TDP-43 in C2C12 cells impairs muscle-characteristic features, like striated muscle development, muscle cell migration or regulation of muscle cell differentiation, which has been functionally confirmed by others[12,29]. However, we have also detected cell-type-specific TDP-43-associated dysregulation of molecular functions that will probably deserve further investigation. For example, we found that in muscles TDP-43 mediates splicing of mRNAs encoding proteins implicated in DNA-related processes. This is a particularly interesting observation as DNA-related processes play an important role in muscle differentiation. In adult skeletal muscle, DNA and histone modifications participate in adaptive response to environmental stimuli, which challenge structural and metabolic demands and thus make skeletal muscle a very plastic tissue[60,61]. Also, the early commitment towards myogenic lineage involves epigenetic changes mediated by chromatin remodelling enzymes like histone deacetylases (HDACs), histone acetyltransferases (HATs) and histone methyltransferases (HMTs)[62]. In keeping with this, *Dnmt3a, Dnmt3b, Hdac9, Hdac7, Prdm2* are just few of chromatin-modifying enzymes that underwent splicing changes upon TDP-43 depletion in C2C12 but not in NSC34 cells. Interestingly, telomere shortening was described in primary muscle cultures of sIBM patients suggesting premature senescence[63] and epigenetic changes have been described in congenital myopathies[64]. Therefore, the results obtained in C2C12 suggest another possible mechanism on how TDP-43 may control gene expression in muscle in an indirect fashion and eventually participate in disease. Recently, loss of TDP-43 was associated with increased genomic instability and R-loop formation[65,66], possibly through mechanisms involving Poldip3, which has been shown to play a role in maintaining genome stability and preventing R-loop accumulation at sites of active replication[67].

On the other hand, some molecular processes such as dysregulation of protein assembly and disposal, mitochondrial changes and apoptosis; as well as alterations in post-translational modifications seem to occur upon TDP-43 depletion in both cell types, which possibly links these pathological changes to TDP-43 dysfunction in both tissues. As we have drawn our conclusions based on the RNA-seq analysis, a crucial future step will be to functionally assess to what extent TDP-43 loss impacts the above-mentioned processes in each tissue. Ideally, functional experiments should be performed in the two cell types in parallel, as only such approach would allow a direct comparison of the regulatory role played by TDP-43 in each context and would answer the question whether impairment of RNA processing is as central in IBM as it is in ALS.

Working with cell lines representing muscles[12,29,48] and neurons[21,23,28,59,68] allowed us a direct (and unbiased) assessment of TDP-43 activity across cel- types. Mouse cell lines have been routinely employed to study TDP-43[12,29]. In our case, they were chosen over human cells due to the lack of an appropriate and well-established cell line derived from human skeletal muscle. With regards to the contribution of TDP-43 malfunction to human pathology, we observed that transcripts, whose splicing was commonly affected by TDP-43 loss in the two mouse cell lines, appear more likely to undergo TDP-43-regulated processing also in human cells. Herein, we show that alternative sequences regulated by TDP-43 in both cell lines are better conserved between species than those regulated in a cell type-specific manner. Nonetheless, a conservation of the alternative sequence itself cannot guarantee for splicing conservation. Thus, it would be extremely insightful to investigate conservation of TDP-43-regulated splicing between humans and mice on a transcriptome-wide level by actual sequencing experiment (rather than comparing gene sequences as such), possibly using analogous tissues[69]. A good example of commonly regulated event is skipping of exon 3 within *POLDIP3*, the regulation of which is conserved between mouse in humans[20,23,32] and has made it the most consistently detected event across studies. In this study, however, we identified two additional targets, *ASAP2* and *PPFIBP1*, and show that they also indeed undergo TDP-43-dependent splicing in all (mouse and human) cell lines tested. These additional findings could be of interest to identify common endpoints of mouse and human disease models that could then be used to monitor the efficiency of eventual therapeutic approaches or to follow disease course/onset.

Finally, as a proof-of-principle, we show that splicing alterations of TDP-43-dependent transcripts do in fact take place in different tissues (i.e., skeletal muscle and certain brain regions) affected by TDP-43 pathology. While expression levels of a given transcript heavily vary between individuals and, in our experience, seem to be influenced by experimental procedure itself (how and when biopsies are taken), the relative abundance of characteristic isoforms appears to be a more reliable readout. Considering cell-type-specific activity of TDP-43, it is reasonable to deduce that splicing of other TDP-43-controlled transcripts would be affected in the skeletal muscle and in neurons. In conclusion, we show that splicing changes represent a robust indication of pathological conditions both in the skeletal muscle of IBM patients and in the brain of individuals affected with FTLD.

## Methods

**Cell culture**. C2C12 immortalized mouse myoblasts (ECACC), SH-SY5Y human neuroblastoma (ECACC) and RH-30 human rhabdomyosarcoma (kindly donated by Marc-David Ruepp) were maintained in DMEM (Thermo Fisher Scientific), supplemented with 10% FBS (Thermo Fisher Scientific) and antibiotics/antimycotics (Sigma-Aldrich) under standard conditions. NSC34 motoneuron-like mouse hybrid cell line (available in house) was cultured in DMEM (Thermo Fisher Scientific) with 5% FBS (Sigma-Aldrich) and antibiotics/antimycotics (Sigma-Aldrich). All experiments were performed with cells of similar passage number (± 2). To silence RNA-binding proteins (TDP-43, Celf2, Khdrbs3, Elavl3, Elavl4) in C2C12 and NSC34 cells, 40 nM of siRNA (mouse siTDP 5'-CGAU-GAACCCAUUGAAAUA-3', Sigma-Aldrich, pre-designed mouse siCelf2, siKhdrbs3, siElavl3 and Elavl4, Thermo Fisher Scientific) or non-targeting siLUC (5'-UAAGGCUAUGAAGAGAUAC-3', Sigma-Aldrich) were mixed with 54 μl of RNAiMAX (Invitrogen) following manufacturer's reverse transfection protocol and applied to 700,000 cells seeded in a 10 cm dish. 48 h later, transfected cells were collected for subsequent analysis. The same reagent was used to silence TDP-43 in human SH-SY5Y and RH-30 cells. 400,000 RH-30 were seeded in a 6 cm dish, reversely transfected (human siTDP 5'-GCAAAGCCAAGAUGAGCCU-3', Sigma-Aldrich or siLUC) and harvested 48 h later. To deplete TDP-43 in SH-SY5Y cells, 10,00,000 cells were seeded in a 6 cm dish and reversely transfected. After 48 h, they were transfected again and harvested 48 h later.

**Western blotting**. Whole-cell extracts were resuspended in PBS in the presence of protease inhibitor and sonicated. 15 μg of protein sample were separated on a 10% Bis-Tris gel (Invitrogen) and transferred to a nitrocellulose membrane (Invitrogen). The membrane was blocked in 4% milk-PBST (PBS with 0.1% Tween-20) and proteins were stained using the following antibodies: anti-TDP-43 (rabbit, Proteintech, 1:1000), anti-GAPDH (rabbit, Proteintech, 1:1000), anti-HSP70 (rat, EnzoLife Science, 1:1000), anti-tubulin (mouse, available in house, 1:10,000) and HRP-conjugated secondary antibodies anti-rabbit (goat, Dako, 1:2000), anti-mouse (goat, Dako, 1:2000), anti-rat (rabbit, Dako, 1:2000).

To detect TBC1D1, 30 μg of protein sample were separated using a pre-cast gradient gel (4–12%, Invitrogen) and wet blotted onto a PVDF membrane (Merck Millipore). The membrane was then blocked in 4% milk-TBST, stained with anti-TBC1D1 (rabbit, Cell Signalling, 1:1 000, in 2% milk-TBST) and anti-rabbit HRP-conjugated secondary antibody (goat, Dako, 1:2 000).

Image acquisition and result quantification were conducted using Alliance Q9 Advanced Chemiluminescence Imager (UviTech).

**RNA extraction, RT-PCR**. Total RNA was isolated using standard phenol-chloroform extraction. RNA quality was assessed using capillary electrophoresis (Qiaxcel RNA QC Kit v2.0, Qiaxcel) and quantified using Qiaxcel software

(QIAxcel ScreenGel (v1.4.0)). Only high-quality (RIS > 8) RNA of high purity ($A_{260}/A_{230}$ and $A_{260}/A_{280}$ > 1.8) was taken for subsequent analysis. 500 ng of RNA were reversely transcribed using random primers (Eurofins) and Moloney murine leukaemia virus reverse transcriptase (M-MLV, Invitrogen) according to manufacturer's instructions.

**Splicing-sensitive PCR and qPCR**. For detection of alternatively spliced mRNAs, PCR primers were designed complementary to constitutive exonic regions flanking a predicted alternatively spliced cassette exon. PCR mix was prepared using gene-specific primers (0.6 μM, Sigma, primer sequences in Supplementary Tables 1 and 2) and TAQ DNA polymerase (Biolabs or Roche) according to manufacturer's instructions and subjected to PCR protocol (35–45 amplification cycles) optimized for each primer pair. PCR products were separated by capillary electrophoresis (DNA screening cartridge, Qiaxcel), and splicing transitions were quantified using Qiaxcel software (Qiaxcel ScreenGel (v1.4.0)). Exon inclusion was calculated by the software. Percentage of the inclusion (Inc. %) reports the area under the curve of the peak representing the longer (inclusion) splicing isoform.

For assessment of transcript levels, real-time quantitative PCR was performed using PowerUp SYBR Green master mix (Applied Biosystems) and gene-specific primers (primer sequences in Supplementary Table 3). cDNA was subjected to 45 cycles of the following thermal protocol: 95 °C for 3 min, 95 °C for 10 s, 65 °C for 30 s, 95 °C for 10 s, 65 °C for 1 s. Relative gene expression levels were determined using QuantStudio design and analysis software (Thermo Fisher Scientific (v1.5.1)) always comparing treated samples (siTDP) with their direct controls (siLUC) normalized against *Gapdh*. *P*-values were calculated using one-tailed paired *t*-test as qPCR was conducted to validate expression changes detected by RNA-seq.

**RNA-seq**. Both polyA cDNA library generation and RNA-seq were performed by Novogene (Beijing, China). cDNA libraries with insert lengths of 250–300 bp were generated using NEB NextUltra RNA Library Prep Kit. Sequencing was conducted on Illumina HiSeq 2500 with paired-end 150 bp (PE 150) strategy.

**Read mapping**. Sequencing quality control and filtering were performed to prune reads with average Phred score (Q score) below 20 across 50% of bases, as well as those with > 0.1% of undetermined (N) ones. Obtained reads were aligned to the mouse genome GRCm38 (mm10) using the Spliced Transcripts Alignment to a Reference (STAR) software (v2.5)[70], an RNA-seq data aligner that utilizes Maximal Mappable Prefix (MMP) strategy to account for the exon junction problem.

**Quantification of gene expression level**. Counting of reads mapped to each gene was performed using HTSeq (v0.6.1)[71]. Raw read counts together with respective gene length were used to calculate Fragments Per Kilobase of transcript sequence per Million base pairs sequenced (FPKM). In contrast to read counts, FPKMs account for sequencing depth and gene length on counting of fragments[72] and are frequently used to estimate gene expression levels.

**Differential expression analysis**. Differential gene expression (DEG) analysis of two conditions was performed using the *DESeq2* R package (v2.1.6.3)[73], a tool that utilizes negative binomial distribution model to account for variance-mean dependence in count data and tests for differential expression[74]. Three biological replicates were included per cell-type and condition, in control (siLUC) and TDP-43-silenced (siTDP) cells. Read count matrix was pre-filtered by removing rows with row sum below 1. Multiple testing adjustments were performed using Benjamini and Hochberg's approach to control for the false discovery rate (FDR). Transcripts with $p_{adj} < 0.05$ were considered as differentially expressed.

Differentially expressed genes identified in both cell lines under different experimental conditions were hierarchically clustered based on $\log_{10}(FPKM + 1)$ and visualized with *pheatmap* R package (v1.0.12)[75]. Further, distance between silenced and control samples of each cell line was illustrated with principal component analysis (PCA), using the R function "*prcomp*"[76]. Differences in gene expression levels ($\log_{10}(FPKM + 1)$) between cell lines were tested for significance using Wilcoxon signed-rank test.

**Alternative splicing analysis**. Five major types of alternative splicing events — skipped exons (SE), mutually exclusive exons (MXE), alternative 5' and 3' splice sites (A5'SS and A3'SS) and intron retention (RI) — were detected and analysed by Novogene using replicate multivariate analysis of transcript splicing (rMATS) software (v3.2.1)[77]. Every alternative splicing event can produce exactly two isoforms. Each isoform is adjusted for its effective length before calculating the ratio of two isoforms and testing significance of differential splicing between two conditions. Multiple testing was corrected using Benjamini and Hochberg's method. Splicing events having FDR < 0.01 were considered significant irrespective of ΔPSI.

Alternatively (for analysis of cryptic splicing and patient's data (Figs. 3c and 9, Supplementary Data 4, Supplementary Figs. 3 and 4a)), differential splicing analysis was performed using MAJIQ (v2.1) and the GRCm38 as a reference genome as previously described elsewhere[78].

**Enrichment analysis**. Gene Ontology GO[79] is widely used in gene enrichment analysis to classify list of individual genes based on their expression pattern, or other similar feature, with the aim to predict dysregulated biological processes, functions and pathways or any other general trend within a subset of data[80]. In this study, GO enrichment analysis was conducted in R, using *clusterProfiler* package (v3.14.3)[80] either on the set of differentially expressed genes ($p_{adj} < 0.05$) or alternatively spliced genes (FDR < 0.01), if not stated otherwise. Additionally, GO enrichment analysis was conducted using a less stringent threshold for inclusion of alternatively spliced genes (where we considered genes with non-corrected *p*-value < 0.01 instead of FDR < 0.01). Genes of a particular dataset were assigned Entrez gene identifiers from Bioconductor mouse annotation package *org.Mm.eg.db* (v3.10.0). Enrichment test for GO terms was calculated based on the hypergeometric distribution. The resulting GO terms/KEGG pathways were considered significant after applying multiple testing corrections with Benjamini–Hochberg method ($p_{adj} < 0.05$). Subsequently, significant GO terms (category: biological process) were functionally grouped or manually edited depending on the underlying biological question.

**Conservation analysis**. Gene/exon conservation analysis within mouse (mm10) was performed by calculating phyloP (phylogenetic *p*-values) scores, i.e., per base conservation scores, generated from aligned genomic sequences of multiple species[81].

For each differentially expressed gene, the average *per gene* phyloP score was computed with bigWigSummary (UCSC)[82]. To calculate phyloP scores of TDP-43-regulated alternative sequences (hereafter referred to as *per exon* phyloP score), we considered TDP-43-regulated sequences of all event types. Those include A'3SS and A'5SS (long and short exon), retained introns and cassette exons (SE, the 1st and the 2nd exon of MXE).

**Patient samples**. The NYGC ALS cohort has previously been detailed elsewhere[22,78]. Herein, we only considered ALS and FTLD patients with TDP-43 pathology (ALS-TDP and FTLD-TDP) and healthy controls while excluding ALS with *SOD1* mutations of FTLD patients without TDP-43 inclusions.

The NYGC ALS Consortium samples presented in this work were acquired through various Institution Review Board protocols from member sites and the Target ALS post-mortem tissue core. They were transferred to the NYGC in accordance with all applicable foreign, domestic, federal, state, and local laws and regulations for processing, sequencing, and analyses.

Muscle biopsies (vastus lateralis or biceps) were obtained from four patients diagnosed with IBM according to the Griggs criteria[83] and four healthy controls. Participants were investigated for cramps or fatigue; they underwent regular examination, neurophysiology tests and histological examinations. IBM biopsies were taken from moderately affected muscles and routinely investigated for histological and immunohistochemistry features. In case muscle fibrosis was present, it did not compromise a definite pathologic diagnosis. Basic demographic features of all participants are summarised in Supplementary Table 4. Biopsies were stored at −80 °C. Institutional board reviewed the study and ethical approval was obtained.

Sample processing, library preparation, and RNA-seq quality control have already been described elsewhere[78].

**Statistics and reproducibility**. Statistical analysis was performed using R (v3.6.1) or a specific R package as described in dedicated sections above (see Quantification of gene expression level, Differential expression analysis, Alternative splicing analysis, Enrichment analysis and Patient samples). A significance level of 0.05 was generally considered for the analysis unless otherwise stated. The exact sample size and statistical tests used to test significance of the difference are given in captions, along with the description of corrections and statistical parameters that are shown in each figure.

At least three biological replicates (experiments conducted at different time points) with technical triplicates were used for qPCR and western blot quantification.

**Reporting summary**. Further information on research design is available in the Nature Research Reporting Summary linked to this article.

## Data availability

Datasets generated for this study were deposited in NCBI's GEO and are accessible through GEO Series accession number GSE171714. iCLIP data generated by Tollervey et al.[21] that support the findings of this study were deposited in the ArrayExpress archive and are accessible at E-MTAB-530. RNA-seq data of the F210I mouse generated by ref. [45] were deposited in the NCBI Sequence Read Archive and are accessible at SRP133158. RNA-seq data generated through the NYGC ALS Consortium used in this study can be accessed at GEO (GSE137810, GSE124439, GSE116622 and GSE153960). To request immediate access to new and ongoing data generated by the NYGC ALS Consortium for samples provided through the Target ALS post-mortem core, contact ALSData@nygenome.org. All RNA-seq data in the NYGC ALS Consortium are made immediately available to all members of the Consortium and with other Consortia with

whom we have a reciprocal sharing arrangement. Uncropped and unedited blot images are included in Supplementary Fig. 6. All data generated for this study is available from the corresponding author on reasonable request.

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

## Acknowledgements

We thank Marc-David Ruepp (King's College London) for providing RH-30 cells and Robert Bakarić for his kind assistance with the conservation analysis. We would also like to thank the Target ALS Human Postmortem Tissue Core (New York Genome Center for Genomics of Neurodegenerative Disease, Amyotrophic Lateral Sclerosis Association) for providing post-mortem brain samples, patients and their families who donated those samples. IBM muscle biopsies were kindly provided by the Bank of muscle tissue, peripheral nerve, DNA and Cell Culture, a member of Telethon Network of Genetic biobanks, at Fondazione IRCCS Ca' Granda, Ospedale Maggiore Policlinico, Milano, Italy and from the Laboratory of Muscle Histopathology and Molecular Biology at IRCCS Policlinico San Donato, San Donato Milanese, Italy. This research was supported by the AriSLA grant PathensTDP to E. Bur. and by the European Reference Network for Neuromuscular Diseases to M. M. and M. Ro. Consortium activities were supported by the ALS Association (15-LGCA-234) and the Tow Foundation. G. M. was supported by Fondazione Malattie Miotoniche, Milan, Italy. Andrea Cortese would like to thank the Medical Research Council (MR/T001712/1), Cariplo Foundation, the Italian Ministry of Health (Ricerca Corrente 2018–2019), the Inherited Neuropathy Consortium and the Fondazione Regionale per la Ricerca Bio-medica for the grant support.

## Author contributions

UŠ and YA conducted experiments; U.Š., N.Š., A.-L.B. and M. Ro. analysed the data; U.Š. and E. Bur. designed the study; H. P. provided patient data collected by the NYGC ALS consortium; A.C., C.C., E. Bug., R.C., G.M., M.Ri., M.M. provided IBM patient samples; P.F., M.S. and E. Bur. supervised the study; U.Š. and E. Bur. wrote the manuscript with contributions of other authors.

## Competing interests

The authors declare no competing interests.

## Additional information

## NYGC ALS Consortium

H. Phatnani[5], P. Fratta[4], J. Kwan[14], D. Sareen[15], J. R. Broach[16], Z. Simmons[17], X. Arcila-Londono[18], E. B. Lee[19], V. M. Van Deerlin[19], N. A. Shneider[20], E. Fraenkel[21], L. W. Ostrow[22], F. Baas[23,24], J. D. Berry[25], O. Butovsky[26], R. H. Baloh[27,28], Ophir Shalem[29,30], T. Heiman-Patterson[31], L. Stefanis[32,33], S. Chandran[34], S. Pal[34], C. Smith[35,36], A. Malaspina[37,38], M. G. Hammell[39], N. A. Patsopoulos[40,41,42], J. Dubnau[43], M. Poss[44], B. Zhang[45], N. Zaitlen[46], E. Hornstein[47], T. M. Miller[48], E. Dardiotis[49], R. Bowser[50], V. Menon[51], M. Harms[52], N. Atassi[53], D. J. Lange[54], D. J. MacGowan[55], C. McMillan[56], E. Aronica[57], B. Harris[58], J. Ravits[59], J. Crary[60], L. M. Thompson[61], T. Raj[62], S. Paganoni[63], D. J. Adams[64,65], S. Babu[66], V. Drory[67], M. Gotkine[68], I. Broce[69], J. Phillips-Cremins[70], A. Nath[71], S. Finkbeiner[72] & G. A. Cox[73]

[14]Department of Neurology, Lewis Katz School of Medicine, Temple University, Philadelphia, PA, USA. [15]Cedars-Sinai Department of Biomedical Sciences, Board of Governors Regenerative Medicine Institute and Brain Program, Cedars-Sinai Medical Center, and Department of Medicine, University of California, Los Angeles, CA, USA. [16]Department of Biochemistry and Molecular Biology, Penn State Institute for Personalized Medicine, The Pennsylvania State University, Hershey, PA, USA. [17]Department of Neurology, The Pennsylvania State University, Hershey, PA, USA. [18]Department of Neurology, Henry Ford Health System, Detroit, MI, USA. [19]Department of Pathology and Laboratory Medicine, Perelman School of Medicine, University of Pennsylvania, Philadelphia, PA, USA. [20]Department of Neurology, Center for Motor Neuron Biology and Disease, Institute for Genomic Medicine, Columbia University, New York, NY, USA. [21]Department of Biological Engineering, Massachusetts Institute of Technology, Cambridge, MA, USA. [22]Department of Neurology, Johns Hopkins School of Medicine, Baltimore, MD, USA. [23]Department of Neurogenetics, Academic Medical Centre, Amsterdam, The Netherlands. [24]Leiden University Medical Center, Leiden, The Netherlands. [25]ALS Multidisciplinary Clinic, Neuromuscular Division, Department of Neurology, Harvard Medical School, and Neurological Clinical Research Institute, Massachusetts General Hospital, Boston, MA, USA. [26]Ann Romney Center for Neurologic Diseases, Brigham and Women's Hospital, Harvard Medical School, Boston, MA, USA. [27]Board of Governors Regenerative Medicine Institute, Los Angeles, CA, USA. [28]Department of Neurology, Cedars-Sinai Medical Center, Los Angeles, CA, USA. [29]Center for Cellular and Molecular Therapeutics, Children's Hospital of Philadelphia, Philadelphia, PA, USA. [30]Department of Genetics, Perelman School of Medicine, University of Pennsylvania, Philadelphia, PA, USA. [31]Center for Neurodegenerative Disorders, Department of Neurology, The Lewis Katz School of Medicine, Temple University, Philadelphia, PA, USA. [32]Center of Clinical Research, Experimental Surgery and Translational Research, Biomedical Research Foundation of the Academy of Athens (BRFAA), Athens, Greece. [33]1st Department of Neurology, Eginition Hospital, Medical School, National and Kapodistrian University of Athens, Athens, Greece. [34]Centre for Clinical Brain Sciences, Anne Rowling Regenerative Neurology Clinic, Euan MacDonald Centre for Motor Neurone Disease Research, University of Edinburgh, Edinburgh, UK. [35]Centre for Clinical Brain Sciences, University of Edinburgh, Edinburgh, UK. [36]Euan MacDonald Centre for Motor Neurone Disease Research, University of Edinburgh, Edinburgh, UK. [37]Centre for Neuroscience and Trauma, Blizard Institute, Barts and The London School of Medicine and Dentistry, Queen Mary University of London, London, UK. [38]Department of Neurology, Basildon University Hospital, Basildon, UK. [39]Cold Spring Harbor Laboratory, Cold Spring Harbor, New York, NY, USA. [40]Computer Science and Systems Biology Program, Ann Romney Center for Neurological Diseases,  Department of Neurology and Division of Genetics in Department of Medicine, Brigham,  UT, USA. [41]Harvard Medical School, Boston, MA, USA. [42]Program in Medical and Population Genetics, Broad Institute, Cambridge, MA, USA. [43]Department of Anesthesiology, Stony Brook University, Stony Brook, NY, USA. [44]Department of Biology and Veterinary and Biomedical Sciences, The Pennsylvania State University, University Park, PA, USA. [45]Department of Genetics and Genomic Sciences, Icahn Institute of Data Science and Genomic Technology, Icahn School of Medicine at Mount Sinai, New York, NY, USA. [46]Department of Medicine, Lung Biology Center, University of California, San Francisco, CA, USA. [47]Department of Molecular Genetics, Weizmann Institute of Science, Rehovot, Israel. [48]Department of Neurology, Washington University in St. Louis, St. Louis, MO, USA. [49]Department of Neurology & Sensory Organs, University of Thessaly, Thessaly, Greece. [50]Department of Neurology, Barrow Neurological Institute, St. Joseph's Hospital and Medical Center, Department of Neurobiology, Barrow Neurological Institute, St. Joseph's Hospital and Medical Center, Phoenix, AZ, USA. [51]Department of Neurology, Columbia University Medical Center, New York, NY, USA. [52]Department of Neurology, Division of Neuromuscular Medicine, Columbia University, New York, NY, USA. [53]Department of Neurology, Harvard Medical School, Neurological Clinical Research Institute, Massachusetts General Hospital, Boston, MA, USA. [54]Department of Neurology, Hospital for Special Surgery and Weill Cornell Medical Center, New York, NY, USA. [55]Department of Neurology, Icahn School of Medicine at Mount Sinai, New York, NY, USA. [56]Department of Neurology, University of Pennsylvania Perelman School of Medicine, Philadelphia, PA, USA. [57]Department of Neuropathology, Academic Medical Center, University of Amsterdam, Amsterdam, The Netherlands. [58]Department of Neuropathology, Georgetown Brain Bank, Georgetown Lombardi Comprehensive Cancer Center, Georgetown University Medical Center, Washington DC, USA. [59]Department of Neuroscience, University of California San Diego, La Jolla, CA, USA. [60]Department of Pathology, Fishberg Department of Neuroscience, Friedman Brain Institute, Ronald M. Loeb Center for Alzheimer's Disease, Icahn School of Medicine at Mount Sinai, New York, NY, USA. [61]Department of Psychiatry & Human Behavior, Department of Biological Chemistry, School of Medicine, and Department of Neurobiology and Behavior, School of Biological Sciences, University California, Irvine, CA, USA. [62]Departments of Neuroscience, and Genetics and Genomic Sciences, Ronald M. Loeb Center for Alzheimer's disease, Icahn School of Medicine at Mount Sinai, New York, NY, USA. [63]Harvard Medical School, Department of Physical Medicine & Rehabilitation, Spaulding Rehabilitation Hospital, Boston, MA, USA. [64]Medical Genetics, Atlantic Health System, Morristown Medical Center, Morristown, NJ, USA. [65]Overlook Medical Center, Summit, NJ, USA. [66]Neurological Clinical Research Institute, Massachusetts General Hospital, Boston, MA, USA. [67]Neuromuscular Diseases Unit, Department of Neurology, Tel Aviv Sourasky Medical Center, Sackler Faculty of Medicine, Tel-Aviv University, Tel-Aviv, Israel. [68]Neuromuscular/EMG service and ALS/Motor Neuron Disease Clinic, Hebrew University-Hadassah Medical Center, Jerusalem, Israel. [69]Neuroradiology Section, Department of Radiology and Biomedical Imaging, University of California, San Francisco, San Francisco, CA, USA. [70]New York Stem Cell Foundation, Department of Bioengineering, School of Engineering and Applied Sciences, University of Pennsylvania, Philadelphia, PA, USA. [71]Section of Infections of the Nervous System, National Institute of Neurological Disorders and Stroke, NIH, Bethesda, MD, USA. [72]Taube/Koret Center for Neurodegenerative Disease Research, Roddenberry Center for Stem Cell Biology and Medicine, Gladstone Institute, San Francisco, CA, USA. [73]The Jackson Laboratory, Bar Harbor, ME, USA.

