## [Peer Review File · Communications Biology]

Reviewers' comments:

Reviewer #1 (Remarks to the Author):

Šušnjar et al. compared transcriptome-wide events controlled by disease-associated TDP-43/TARDBP protein in mouse cells of muscle (C2C12) and neural (NSC34) lineages. The authors show that, although TDP-43 is expressed at comparable levels in the two lineages, its knockdown leads to clearly distinct, cell type-specific changes in gene expression and alternative splicing patterns. Functional annotation of genes regulated by TDP43 reveals enrichment in lineage-specific categories but also common processes that might contribute to pathologies in both muscle and brain. Splicing events controlled by TDP-43 are more dependent on the cell type with "neuronal" GO terms enriched in NSC34 and, surprisingly, "DNA-related" ones, in C2C12. Evidence is provided that the lineage-specific effects of TDP-43 on splicing are modulated by cell type-specific RNA-binding proteins including Elavl3/HuC and Elavl4/HuD in NSC34 and Khdrbs2/Slm1 and Celf2/Cugbp2 in C2C12. By comparing RNA-seq data from the two cell types the authors have identified several new alternative-splicing events that are likely controlled by TDP-43 directly. At least some of these events are conserved between mouse and human and are deregulated in the context of neurodegenerative (ALS/FTLD) and muscle (IBM) diseases extending the range of molecular markers for research and diagnostic purposes.

Although TDP-43 has been extensively studied in brain, its functions in muscle cells and their progenitors have not been understood to the same extent. The study by Šušnjar et al. fills in this important gap. Their work also provides high-quality comparative transcriptome-wide data that will benefit researchers interested in TDP-43, its role in pathologies, and wider aspects of RNA-based regulation of gene expression in health and disease. Numbers of repeats and statistical analyses of the data seem appropriate. I therefore think that the manuscript is sufficiently novel and interesting to be considered for publication. My suggestions for improvement are as follows.

1. The current version of the manuscript provides no insights into molecular mechanism(s) that might allow TDP-43 to control abundance of hundreds and possibly thousands of cell-type specific transcripts. Fig S3A showing overlap between DEG and AS targets is interesting in this respect since looks like AS-controlled genes might be enriched for the DEG behaviour. The authors should analyse these Venn diagrams by Fisher's exact test to see if this effect is significant. NMD is mentioned as a possible link between AS and DEG in the text, but no further details are provided. I suggest exploring this possibility systematically using appropriate bioinformatics tools (e.g. <https://fursham-h.github.io/factR/articles/factR.html>). It would really strengthen the paper if the authors could explain how TDP-43 controls abundance of at least a subset of transcripts.
2. Have the authors compared their NSC34 data to published RNA-seq studies where TDP-43 was inactivated in neuronal cells? A significant overlap of TDP-43 targets between cells belonging to the same lineage would strengthen authors' argument about lineage specificity of TDP-43 regulation.
3. Throughout the manuscript, "transcription" and "transcriptional" are used to describe specific RNAs expression patterns (lines 130, 132, 137, etc.). This is misleading since transcription is only one of several processes controlling the abundance of mature RNAs in the cell. The authors should consider alternative terminology, for example "transcriptome", "gene expression" or "RNA abundance".
4. Line 159: Wrong Greek letter - phi instead of rho.
5. Lines 162-163: I would avoid the loss-of-function/gain-of-function terminology in this context. It is better suited for classifying mutations rather than effects of protein knockdown.
6. Line 334: The claim that alternative splicing occurs co-transcriptionally should be toned down. Not all splicing events occur co-transcriptionally.
7. Finally, all abbreviations should be spelled out when using them for the first time (e.g. what do FTLD and IBM stand for in the Abstract?). Also, FTLD is technically "frontotemporal lobar

degeneration", not "frontotemporal dementia" (FTD).

Reviewer #2 (Remarks to the Author):

In this study, Susnjar and colleagues set out to investigate the role of TDP-43 in gene expression beyond the nervous system. This is an important field of investigation since TDP-43 inclusions are also found outside of neuronal tissues, implicating that TDP-43 dysfunction might be deleterious for other tissues and could contribute to pathogenesis.

To address this question, the authors performed an RNA-Seq based comparison between the mouse motor neuron-like cell line NSC-34 and the mouse myoblast cell line C2C12 in the presence or absence of TDP-43 depletion. This approach allowed them to gain insights into the function of TDP-43 in neuron-like and muscle-like cells. Their results show that TDP-43 depletion results in not only shared but also cell-type specific changes in gene expression implicating that TDP-43 dysfunction results in tissue specific changes. The cell-specific changes seem to be a consequence of cell-specific levels of RBPs that co-modulate RNA processing events together with TDP-43. They then validated a subset of splicing events in human cells as well as in FTLD and IBM patient samples, linking identified events in their cell system to disease.

The manuscript is well written and gives us first insights into cell-type specific control of gene expression by TDP-43 with important implications for our understanding of how TDP-43 functions in health and disease. The study also highlights the importance of cellular context for pathomechanistic research. I enjoyed reading the manuscript and only have minor comments and questions.

Comments/queries:

41: Can one directly infer tissue characteristic behaviour from cell lines? Or wouldn't "cell-type characteristic behaviour" be more appropriate which then implicates tissue characteristic behaviour?

128: Did the authors mean "deep transcriptome analysis by RNA-Seq" instead of "deep RNA-Seq"?

554: Does the T in PBST stand for Tween-20, and could the authors give the percentage of Tween used?

563: Could the authors mention how RIS was assessed?

590: Could the authors state which Illumina platform was used?

Typos/Grammar queries:

553: "blocked" instead of "bloctfisherked"?

562: "high quality" instead of "undegraded"?

556: "in house" instead of "inhouse"?

564: "500 ng of RNA were" instead of "was"?

573: "35-45 amplification cycles" instead of "cycles long thermal protocol"?

584: "P-values" instead of "p-values"

589: "insert lengths" instead of "insert length"?

677: "stored at -80" instead of "80"?

Colours:

I really had problems differentiating the chosen colours for the datasets in figures 2D, 2F, 5B. I would suggest a change in colour palette to make it easier for the reader.

Experimental suggestion:

As the authors state, this work is mainly based on RNA-Seq and functional assessment will be a crucial future step. While functional assays will be outside the scope of this manuscript this reviewer wonders whether the manuscript would not profit from the validation of some of the identified changes on the protein level by western blot or IF. For example, a blot using mouse or human cell lysates showing the impact of TDP-43 depletion for some proteins/isoforms that are

generally affected and affected in a cell-type-specific manner since RNA levels do not necessarily directly relate to protein levels.

Reviewer #1 (Remarks to the Author):

Šušnjar et al. compared transcriptome-wide events controlled by disease-associated TDP-43/TARDBP protein in mouse cells of muscle (C2C12) and neural (NSC34) lineages. The authors show that, although TDP-43 is expressed at comparable levels in the two lineages, its knockdown leads to clearly distinct, cell type-specific changes in gene expression and alternative splicing patterns. Functional annotation of genes regulated by TDP43 reveals enrichment in lineage-specific categories but also common processes that might contribute to pathologies in both muscle and brain. Splicing events controlled by TDP-43 are more dependent on the cell type with "neuronal" GO terms enriched in NSC34 and, surprisingly, "DNA-related" ones, in C2C12. Evidence is provided that the lineage-specific effects of TDP-43 on splicing are modulated by cell type-specific RNA-binding proteins including Elavl3/HuC and Elavl4/HuD in NSC34 and Khdrbs2/Slm1 and Celf2/Cugbp2 in C2C12. By comparing RNA-seq data from the two cell types the authors have identified several new alternative-splicing events that are likely controlled by TDP-43 directly. At least some of these events are conserved between mouse and human and are deregulated in the context of neurodegenerative (ALS/FTLD) and muscle (IBM) diseases extending the range of molecular markers for research and diagnostic purposes.

Although TDP-43 has been extensively studied in brain, its functions in muscle cells and their progenitors have not been understood to the same extent. The study by Šušnjar et al. fills in this important gap. Their work also provides high-quality comparative transcriptome-wide data that will benefit researchers interested in TDP-43, its role in pathologies, and wider aspects of RNA-based regulation of gene expression in health and disease. Numbers of repeats and statistical analyses of the data seem appropriate. I therefore think that the manuscript is sufficiently novel and interesting to be considered for publication. My suggestions for improvement are as follows.

1. The current version of the manuscript provides no insights into molecular mechanism(s) that might allow TDP-43 to control abundance of hundreds and possibly thousands of cell-type specific transcripts. Fig S3A showing overlap between DEG and AS targets is interesting in this respect since looks like AS-controlled genes might be enriched for the DEG behaviour. The authors should analyse these Venn diagrams by Fisher's exact test to see if this effect is significant. NMD is mentioned as a possible link between AS and DEG in the text, but no further details are provided. I suggest exploring this possibility systematically using appropriate bioinformatics tools (e.g. <https://fursham-h.github.io/factR/articles/factR.html>). It would really strengthen the paper if the authors could explain how TDP-43 controls abundance of at least a subset of transcripts.

ANSWER: The question regarding possible mechanistic explanation on how TDP-43 controls abundance of its targets has been partially answered by results obtained by Polymenidou *et al.* (2011) and Cortese *et al.* (2014). More specifically, Polymenidou *et al.*, (2011) suggested TDP-43 binding to be crucial in sustaining pre-mRNA levels and that mRNA downregulation occurs as a direct consequence of TDP-43 loss, while mRNA upregulation was attributed to indirect effects. In neurons, TDP-43 binding sites were found in protein-coding genes with particularly long introns. In line with that, Cortese *et al.* (2014) additionally demonstrated

that transcripts with long introns are significantly downregulated in sporadic inclusion body myositis (sIBM). Despite not being discussed in this paper, we have seen that genes targeted by TDP-43 in C2C12 and NSC34 are significantly longer ($p\text{-value} < 2.2 \cdot 10^{-16}$) than an average gene in mouse genome (see the figure below). What is more, TDP-43-controlled genes in NSC34 cells appear to be longer than TDP-43 targets in C2C12 ($p\text{-value} = 2.1 \cdot 10^{-7}$). Long genes are expected to contain exceptionally long introns (but not exons) which hold important regulatory elements including those bound by TDP-43.

Gene length of TDP-43 targets in C2C12 and NSC34. A Cumulative distribution plot compares gene lengths of TDP-43-targeted genes (2325 and 2324 in C2C12 and NSC34, respectively) with other genes in mouse genome (38,793). TDP-43-controlled genes are significantly longer ($p\text{-value} < 2.2 \cdot 10^{-16}$) than other genes of mouse genome, while DEGs detected in NSC34 appear to be longer than those regulated by TDP-43 in C2C12 ($p\text{-value} = 2.1 \cdot 10^{-7}$).

We have indeed hypothesized that TDP-43 loss induced changes in transcript abundance might result as a consequence of AS and consequently, NMD, however, this hypothesis was not thoroughly elaborated. We have now tested significance of the overlap between DEG and AS genes identified in each cell line using Fisher's exact test, as suggested by the reviewer, and it turns out those overlaps are in fact not significant ($p\text{-values}$ of 0.06 and 0.19 in C2C12 and NSC34, respectively). Therefore, we would like to exclude the part discussing the link between TDP-43-mediated AS and DEG from the revised version of the manuscript.

As a result, Supplementary Fig.3 was removed from the manuscript and was replaced by the comparison requested by the reviewer and explained in the subsequent point.

2. Have the authors compared their NSC34 data to published RNA-seq studies where TDP-43 was inactivated in neuronal cells? A significant overlap of TDP-43 targets between cells belonging to the same lineage would strengthen authors' argument about lineage specificity of TDP-43 regulation.

Appendix figure S3. **TDP-43-regulated splicing changes in mouse cell lines and bulk brain tissue.**

Venn diagram shows the number of AS events (junctions) detected by MAJIQ ($\Delta\text{PSI} > 0.2$, $\text{FDR} < 0.1$) detected in C2C12 and NSC34 cell lines (as in **Fig 3B**) together with those identified in brain tissue of mice carrying a mutation in the RNA-recognition motif RRM2 (F210I) of endogenous *Tardbp*. That RNA-seq dataset was generated by Fratta *et al* (2018).

ANSWER: As suggested by the reviewer, we compared results obtained in TDP-43-silenced cells (NSC34 and C2C12) to a RNA-seq dataset previously generated by Fratta *et al* (2018), in which they sequenced brain tissue of mice carrying a mutation in important RNA-recognition motif RRM2 (F210I) of endogenous *Tardbp* gene leading to partial loss of TDP-43's splicing function.

That dataset was subjected to the same MAJIQ pipeline for detection of alternatively used splicing junctions (**Fig. 3C**). The overlap between splicing events detected in TDP-43-silenced NSC34 cells and brain tissue of mice with the RRM2 mutation was 10.1 % compared to 15.2% of splicing events commonly identified in C2C12 and NSC34 following TDP-43 knockdown. This may seem rather low. However, several considerations must be made: first, cell lines (C2C12 or NSC34, respectively) consist of homogenous cell populations, which is in contrast to heterogeneous cell composition within bulk brain tissue. Second, different from embryonic mouse brain (E18.5) containing neuronal cells at different stages of differentiation, only undifferentiated cells were used in our study (C2C12 and NSC34). It should also not be neglected that TDP-43 depletion in C2C12 and NSC34 cells was achieved by siRNA transfection whilst there was some functional TDP-43 still present in the F210I embryonic brain (as complete knockout of TDP-43 in mouse embryo leads to a very early embryonic lethality before before day 12 of embryogenesis (Kraemer *et al* 2010).

For these reasons, we believe that results generated by a comparative analysis of TDP-43's function in NSC34 and C2C12 cells clearly demonstrate that the activity of TDP-43 itself depends on the cellular context. Most importantly, we believe that this 10.1% overlap consistently supports the idea that there is a small subset of transcript targeted by TDP-43 across different cell types (like for example *Poldip3*, *Ppfibp1*, *Tmem2*) while splicing of others strongly depends on cell-specific factors.

3. Throughout the manuscript, “transcription” and “transcriptional” are used to describe specific RNAs expression patterns (lines 130, 132, 137, etc.). This is misleading since transcription is only one of several processes controlling the abundance of mature RNAs in the cell. The authors should consider alternative terminology, for example “transcriptome”, “gene expression” or “RNA abundance”.

ANSWER: Corrected. Changes are marked in red.

4. Line 159: Wrong Greek letter - phi instead of rho.

ANSWER: Corrected. The change is marked in red.

5. Lines 162-163: I would avoid the loss-of-function/gain-of-function terminology in this context. It is better suited for classifying mutations rather than effects of protein knockdown.

ANSWER: These terms were left out.

6. Line 334: The claim that alternative splicing occurs co-transcriptionally should be toned down. Not all splicing events occur co-transcriptionally.

ANSWER: We certainly agree with the reviewer. In fact, we had the intention to point out the two mechanisms could be linked (in certain cases) rather than claim that splicing occurs co-transcriptionally or that the two mechanisms would be connected in a specific event identified in this study.

After having seen that the overlap between TDP-43-controlled DE and AS transcripts was not significant (referring to question nr. 1) we would like to tone down this aspect of the manuscript and exclude that part (i.e., discussion whether AS and DEG are linked) from the revised version. We believe this question deserves some attention but would need to be addressed by combining the bioinformatic analysis (of this dataset) and appropriate experiments.

7. Finally, all abbreviations should be spelled out when using them for the first time (e.g. what do FTLD and IBM stand for in the Abstract?). Also, FTLD is technically "frontotemporal lobar degeneration", not "frontotemporal dementia" (FTD).

ANSWER: Added and corrected. Changes are marked in red.

Reviewer #2 (Remarks to the Author):

In this study, Susnjar and colleagues set out to investigate the role of TDP-43 in gene expression beyond the nervous system. This is an important field of investigation since TDP-43 inclusions are also found outside of neuronal tissues, implicating that TDP-43 dysfunction might be deleterious for other tissues and could contribute to pathogenesis.

To address this question, the authors performed an RNA-Seq based comparison between the mouse motor neuron-like cell line NSC-34 and the mouse myoblast cell line C2C12 in the presence or absence of TDP-43 depletion. This approach allowed them to gain insights into the function of TDP-43 in neuron-like and muscle-like cells. Their results show that TDP-43 depletion results in not only shared but also cell-type specific changes in gene expression implicating that TDP-43 dysfunction results in tissue specific changes. The cell-specific changes seem to be a consequence of cell-specific levels of RBPs that co-modulate RNA processing events together with TDP-43. They then validated a subset of splicing events in human cells as well as in FTLD and IBM patient samples, linking identified events in their cell system to disease.

The manuscript is well written and gives us first insights into cell-type specific control of gene expression by TDP-43 with important implications for our understanding of how TDP-43 functions in health and disease. The study also highlights the importance of cellular context for pathomechanistic research. I enjoyed reading the manuscript and only have minor comments and questions.

Comments/queries:

41: Can one directly infer tissue characteristic behaviour from cell lines? Or wouldn't "cell-type characteristic behaviour" be more appropriate which then implicates tissue characteristic behaviour?

ANSWER: TDP-43-dependent differences in RNA processing described herein serve as a proof-of-principle, implying TDP-43's activity probably differs between tissues. However, we agree with the reviewer – our findings obtained using undifferentiated cell lines can recapitulate only to some extent the tissue-specific behaviour (for a better explanation of this point, please see our answer to question 2 of reviewer number 1 and our inclusion of a new supplementary figure). Therefore, as suggested, in this revised version we have now used the term "cell-type characteristic" where this was more appropriate. All changes are marked in red.

128: Did the authors mean "deep transcriptome analysis by RNA-Seq" instead of "deep RNA-Seq"?

ANSWER: Yes. We intended to say "RNA-Seq with high read coverage". We could simply use the term "RNA-Seq". See the correction in red.

554: Does the T in PBST stand for Tween-20, and could the authors give the percentage of Tween used?

ANSWER: Yes, we used PBS with 0.1% Tween-20. This information was added, see it in red.

563: Could the authors mention how RIS was assessed?

ANSWER: RNA quality was assessed using capillary electrophoresis (QIAxcel RNA QC Kit v2.0, Qiaxcel) and quantified using Qiaxcel software (QIAxcel ScreenGel (v1.4.0)).

This information was added, please see it in red.

590: Could the authors state which Illumina platform was used?

ANSWER: Yes, it was HiSeq 2500. This information was added.

Typos/Grammar queries:

553: “blocked” instead of “bloctfisherked”? ANSWER: Corrected.

562: “high quality” instead of “undegraded”? ANSWER: Corrected.

556: “in house” instead of “inhouse”? ANSWER: Corrected.

564: “500 ng of RNA were” instead of “was”? ANSWER: Corrected.

573: “35-45 amplification cycles” instead of “cycles long thermal protocol”? ANSWER: Corrected.

584: “P-values” instead of “p-values” ANSWER: Corrected.

589: “insert lengths” instead of “insert length”? ANSWER: Corrected.

677: “stored at -80” instead of “80”? ANSWER: Corrected.

Colours:

I really had problems differentiating the chosen colours for the datasets in figures 2D, 2F, 5B. I would suggest a change in colour palette to make it easier for the reader.

ANSWER: The same colour palette is used for differentially expressed genes (red for C2C12 and blue for NSC34) throughout the paper, therefore, we would not like to use different colours for these figures specifically. As suggested by the reviewer, however, we tried to improve visibility using darker shades of the same colour (red and blue). Having tested different colour combinations, we think figures are difficult to read is due to the small size of individual datapoints and small font of the text rather than the colour itself.

Experimental suggestion:

As the authors state, this work is mainly based on RNA-Seq and functional assessment will be a crucial future step. While functional assays will be outside the scope of this manuscript this reviewer wonders whether the manuscript would not profit from the validation of some of the identified changes on the protein level by western blot or IF. For example, a blot using mouse or human cell lysates showing the impact of TDP-43 depletion for some proteins/isoforms that are generally affected and affected in a cell-type-specific manner since RNA levels do not necessarily directly relate to protein levels.

ANSWER: The reviewer is certainly correct in asking for validation of some results by Western blots. This is an ongoing work in the lab. Of course, for many proteins, there is the

difficulty of finding antibodies specific for the various isoforms and/or capable of detecting them. To comprehensively answer this question, this would require an extensive screening using isoform-specific antibodies or a mass-spectrometry-based approach. However, after several tries using commercial antibodies able in principle to see some of the isoforms we have quantified at the mRNA level, we have been successful for one of the C2C12-specific splicing events found in this work, namely TBC1D1, where the commercial antibody can recognize both, the long and the short isoform. We are now commenting on this issue in Discussion and present this data as a new supplementary figure (**Appendix Fig S5**).

In this figure, as a *proof-of-principle*, we show that for TBC1D1, TDP-43-dependent ratio of two protein isoforms actually reflects that of *Tbc1d1* mRNAs (**Appendix Fig S5**):

Appendix figure S5. **TDP-43 dependent expression of TBC1D1 isoforms.**

As shown in this figure, both *Tbc1d1* mRNA isoforms detected in C2C12 cells (**Fig 6B**) are translated into proteins. Western blot shows expression of both TBC1D1 protein isoforms (128 and 138 kDa, respectively) in siLUC-transfected or untreated (ctrl.) C2C12 cells, while only the smaller protein isoform is expressed in TDP-43-silenced cells. The amount of TBC1D1 was normalized against GAPDH or tubulin).